# Efficient Globally Convergent Stochastic Optimization for Canonical Correlation Analysis

**Weiran Wang**[1*]     **Jialei Wang**[2*]     **Dan Garber**[1]     **Nathan Srebro**[1]
[1]Toyota Technological Institute at Chicago     [2]University of Chicago
{weiranwang,dgarber,nati}@ttic.edu     jialei@uchicago.edu

## Abstract

We study the stochastic optimization of canonical correlation analysis (CCA), whose objective is nonconvex and does not decouple over training samples. Although several stochastic gradient based optimization algorithms have been recently proposed to solve this problem, no global convergence guarantee was provided by any of them. Inspired by the alternating least squares/power iterations formulation of CCA, and the shift-and-invert preconditioning method for PCA, we propose two globally convergent meta-algorithms for CCA, both of which transform the original problem into sequences of least squares problems that need only be solved approximately. We instantiate the meta-algorithms with state-of-the-art SGD methods and obtain time complexities that significantly improve upon that of previous work. Experimental results demonstrate their superior performance.

## 1 Introduction

Canonical correlation analysis (CCA, [1]) and its extensions are ubiquitous techniques in scientific research areas for revealing the common sources of variability in multiple views of the same phenomenon. In CCA, the training set consists of paired observations from two views, denoted $(\mathbf{x}_1, \mathbf{y}_1), \ldots, (\mathbf{x}_N, \mathbf{y}_N)$, where $N$ is the training set size, $\mathbf{x}_i \in \mathbb{R}^{d_x}$ and $\mathbf{y}_i \in \mathbb{R}^{d_y}$ for $i = 1, \ldots, N$. We also denote the data matrices for each view[2] by $\mathbf{X} = [\mathbf{x}_1, \ldots, \mathbf{x}_N] \in \mathbb{R}^{d_x \times N}$ and $\mathbf{Y} = [\mathbf{y}_1, \ldots, \mathbf{y}_N] \in \mathbb{R}^{d_y \times N}$, and $d := d_x + d_y$. The objective of CCA is to find linear projections of each view such that the correlation between the projections is maximized:

$$\max_{\mathbf{u}, \mathbf{v}} \quad \mathbf{u}^\top \boldsymbol{\Sigma}_{xy} \mathbf{v} \qquad \text{s.t.} \quad \mathbf{u}^\top \boldsymbol{\Sigma}_{xx} \mathbf{u} = \mathbf{v}^\top \boldsymbol{\Sigma}_{yy} \mathbf{v} = 1 \tag{1}$$

where $\boldsymbol{\Sigma}_{xy} = \frac{1}{N} \mathbf{X} \mathbf{Y}^\top$ is the cross-covariance matrix, $\boldsymbol{\Sigma}_{xx} = \frac{1}{N} \mathbf{X} \mathbf{X}^\top + \gamma_x \mathbf{I}$ and $\boldsymbol{\Sigma}_{yy} = \frac{1}{N} \mathbf{Y} \mathbf{Y}^\top + \gamma_y \mathbf{I}$ are the auto-covariance matrices, and $(\gamma_x, \gamma_y) \geq 0$ are regularization parameters [2].

We denote by $(\mathbf{u}^*, \mathbf{v}^*)$ the global optimum of (1), which can be computed in closed-form. Define

$$\mathbf{T} := \boldsymbol{\Sigma}_{xx}^{-\frac{1}{2}} \boldsymbol{\Sigma}_{xy} \boldsymbol{\Sigma}_{yy}^{-\frac{1}{2}} \ \in \mathbb{R}^{d_x \times d_y}, \tag{2}$$

and let $(\boldsymbol{\phi}, \boldsymbol{\psi})$ be the (unit-length) left and right singular vector pair associated with $\mathbf{T}$'s largest singular value $\rho_1$. Then the optimal objective value, i.e., the canonical correlation between the views, is $\rho_1$, achieved by $(\mathbf{u}^*, \mathbf{v}^*) = (\boldsymbol{\Sigma}_{xx}^{-\frac{1}{2}} \boldsymbol{\phi}, \boldsymbol{\Sigma}_{yy}^{-\frac{1}{2}} \boldsymbol{\psi})$. Note that

$$\rho_1 = \|\mathbf{T}\| \leq \left\| \boldsymbol{\Sigma}_{xx}^{-\frac{1}{2}} \mathbf{X} \right\| \left\| \boldsymbol{\Sigma}_{yy}^{-\frac{1}{2}} \mathbf{Y} \right\| \leq 1.$$

Furthermore, we are guaranteed to have $\rho_1 < 1$ if $(\gamma_x, \gamma_y) > 0$.

---

[*]The first two authors contributed equally.

[2]We assume that $\mathbf{X}$ and $\mathbf{Y}$ are centered at the origin for notational simplicity; if they are not, we can center them as a pre-processing operation.

Table 1: Time complexities of different algorithms for achieving $\eta$-suboptimal solution $(\mathbf{u}, \mathbf{v})$ to CCA, i.e., $\min\left((\mathbf{u}^\top \boldsymbol{\Sigma}_{xx} \mathbf{u}^*)^2, (\mathbf{v}^\top \boldsymbol{\Sigma}_{yy} \mathbf{v}^*)^2\right) \geq 1 - \eta$. GD=gradient descent, AGD=accelerated GD, SVRG=stochastic variance reduced gradient, ASVRG=accelerated SVRG. Note ASVRG provides speedup over SVRG only when $\tilde{\kappa} > N$, and we show the dominant term in its complexity.

| Algorithm | Least squares solver | Time complexity |
|---|---|---|
| `AppGrad` [3] | GD | $\tilde{\mathcal{O}}\left(dN\tilde{\kappa}\frac{\rho_1^2}{\rho_1^2-\rho_2^2}\cdot\log\left(\frac{1}{\eta}\right)\right)$     (local) |
| `CCALin` [6] | AGD | $\tilde{\mathcal{O}}\left(dN\sqrt{\tilde{\kappa}}\frac{\rho_1^2}{\rho_1^2-\rho_2^2}\cdot\log\left(\frac{1}{\eta}\right)\right)$ |
| This work: Alternating least squares (ALS) | AGD | $\tilde{\mathcal{O}}\left(dN\sqrt{\tilde{\kappa}}\left(\frac{\rho_1^2}{\rho_1^2-\rho_2^2}\right)^2\cdot\log^2\left(\frac{1}{\eta}\right)\right)$ |
| | SVRG | $\tilde{\mathcal{O}}\left(d(N+\tilde{\kappa})\left(\frac{\rho_1^2}{\rho_1^2-\rho_2^2}\right)^2\cdot\log^2\left(\frac{1}{\eta}\right)\right)$ |
| | ASVRG | $\tilde{\mathcal{O}}\left(d\sqrt{N\tilde{\kappa}}\left(\frac{\rho_1^2}{\rho_1^2-\rho_2^2}\right)^2\cdot\log^2\left(\frac{1}{\eta}\right)\right)$ |
| This work: Shift-and-invert preconditioning (SI) | AGD | $\tilde{\mathcal{O}}\left(dN\sqrt{\tilde{\kappa}}\sqrt{\frac{1}{\rho_1-\rho_2}}\cdot\log^2\left(\frac{1}{\eta}\right)\right)$ |
| | SVRG | $\tilde{\mathcal{O}}\left(d\left(N+(\tilde{\kappa}\frac{1}{\rho_1-\rho_2})^2\right)\cdot\log^2\left(\frac{1}{\eta}\right)\right)$ |
| | ASVRG | $\tilde{\mathcal{O}}\left(dN^{\frac{3}{4}}\sqrt{\tilde{\kappa}}\sqrt{\frac{1}{\rho_1-\rho_2}}\cdot\log^2\left(\frac{1}{\eta}\right)\right)$ |

For large and high dimensional datasets, it is time and memory consuming to first explicitly form the matrix $\mathbf{T}$ (which requires eigen-decomposition of the covariance matrices) and then compute its singular value decomposition (SVD). For such datasets, it is desirable to develop stochastic algorithms that have efficient updates, converges fast, and takes advantage of the input sparsity. There have been recent attempts to solve (1) based on stochastic gradient descent (SGD) methods [3, 4, 5], but none of these work provides rigorous convergence analysis for their stochastic CCA algorithms.

The main contribution of this paper is the proposal of two globally convergent meta-algorithms for solving (1), namely, alternating least squares (ALS, Algorithm 2) and shift-and-invert preconditioning (SI, Algorithm 3), both of which transform the original problem (1) into sequences of least squares problems that need only be solved approximately. We instantiate the meta algorithms with state-of-the-art SGD methods and obtain efficient stochastic optimization algorithms for CCA.

In order to measure the alignments between an approximate solution $(\mathbf{u}, \mathbf{v})$ and the optimum $(\mathbf{u}^*, \mathbf{v}^*)$, we assume that $\mathbf{T}$ has a positive singular value gap $\Delta := \rho_1 - \rho_2 \in (0, 1]$ so its top left and right singular vector pair is unique (up to a change of sign).

Table 1 summarizes the time complexities of several algorithms for achieving $\eta$-suboptimal alignments, where $\tilde{\kappa} = \frac{\max\limits_{i} \max\left(\|\mathbf{x}_i\|^2, \|\mathbf{y}_i\|^2\right)}{\min(\sigma_{\min}(\boldsymbol{\Sigma}_{xx}), \sigma_{\min}(\boldsymbol{\Sigma}_{yy}))}$ is the upper bound of condition numbers of least squares problems solved in all cases.[3] We use the notation $\tilde{\mathcal{O}}(\cdot)$ to hide poly-logarithmic dependencies (see Sec. 3.1.1 and Sec. 3.2.3 for the hidden factors). Each time complexity may be preferrable in certain regime depending on the parameters of the problem.

**Notations** We use $\sigma_i(\mathbf{A})$ to denote the $i$-th largest singular value of a matrix $\mathbf{A}$, and use $\sigma_{\max}(\mathbf{A})$ and $\sigma_{\min}(\mathbf{A})$ to denote the largest and smallest singular values of $\mathbf{A}$ respectively.

## 2  Motivation: Alternating least squares

Our solution to (1) is inspired by the alternating least squares (ALS) formulation of CCA [7, Algorithm 5.2], as shown in Algorithm 1. Let the nonzero singular values of $\mathbf{T}$ be $1 \geq \rho_1 \geq \rho_2 \geq \cdots \geq \rho_r > 0$, where $r = \text{rank}(\mathbf{T}) \leq \min(d_x, d_y)$, and the corresponding (unit-length) left and right singular vector pairs be $(\mathbf{a}_1, \mathbf{b}_1), \ldots, (\mathbf{a}_r, \mathbf{b}_r)$, with $\mathbf{a}_1 = \phi$ and $\mathbf{b}_1 = \psi$. Define

$$\mathbf{C} = \left[\begin{array}{cc} \mathbf{0} & \mathbf{T} \\ \mathbf{T}^\top & \mathbf{0} \end{array}\right] \in \mathbb{R}^{d \times d}. \tag{3}$$

**Algorithm 1** Alternating least squares for CCA.

---

**Input:** Data matrices $\mathbf{X} \in \mathbb{R}^{d_x \times N}$, $\mathbf{Y} \in \mathbb{R}^{d_y \times N}$, regularization parameters $(\gamma_x, \gamma_y)$.

Initialize $\tilde{\mathbf{u}}_0 \in \mathbb{R}^{d_x}$, $\quad \tilde{\mathbf{v}}_0 \in \mathbb{R}^{d_y}$. $\hspace{4cm} \left\{ \tilde{\phi}_0, \ \tilde{\psi}_0 \right\}$

$\mathbf{u}_0 \leftarrow \tilde{\mathbf{u}}_0 / \sqrt{\tilde{\mathbf{u}}_0^\top \mathbf{\Sigma}_{xx} \tilde{\mathbf{u}}_0}, \quad \mathbf{v}_0 \leftarrow \tilde{\mathbf{v}}_0 / \sqrt{\tilde{\mathbf{v}}_0^\top \mathbf{\Sigma}_{yy} \tilde{\mathbf{v}}_0} \quad \left\{ \phi_0 \leftarrow \tilde{\phi}_0 / \left\| \tilde{\phi}_0 \right\|, \ \psi_0 \leftarrow \tilde{\psi}_0 / \left\| \tilde{\psi}_0 \right\| \right\}$

**for** $t = 1, 2, \ldots, T$ **do**

$\quad \tilde{\mathbf{u}}_t \leftarrow \mathbf{\Sigma}_{xx}^{-1} \mathbf{\Sigma}_{xy} \mathbf{v}_{t-1} \hspace{4cm} \left\{ \tilde{\phi}_t \leftarrow \mathbf{\Sigma}_{xx}^{-\frac{1}{2}} \mathbf{\Sigma}_{xy} \mathbf{\Sigma}_{yy}^{-\frac{1}{2}} \psi_{t-1} \right\}$

$\quad \tilde{\mathbf{v}}_t \leftarrow \mathbf{\Sigma}_{yy}^{-1} \mathbf{\Sigma}_{xy}^\top \mathbf{u}_{t-1} \hspace{4cm} \left\{ \tilde{\psi}_t \leftarrow \mathbf{\Sigma}_{yy}^{-\frac{1}{2}} \mathbf{\Sigma}_{xy}^\top \mathbf{\Sigma}_{xx}^{-\frac{1}{2}} \phi_{t-1} \right\}$

$\quad \mathbf{u}_t \leftarrow \tilde{\mathbf{u}}_t / \sqrt{\tilde{\mathbf{u}}_t^\top \mathbf{\Sigma}_{xx} \tilde{\mathbf{u}}_t}, \quad \mathbf{v}_t \leftarrow \tilde{\mathbf{v}}_t / \sqrt{\tilde{\mathbf{v}}_t^\top \mathbf{\Sigma}_{yy} \tilde{\mathbf{v}}_t} \quad \left\{ \phi_t \leftarrow \tilde{\phi}_t / \left\| \tilde{\phi}_t \right\|, \ \psi_t \leftarrow \tilde{\psi}_t / \left\| \tilde{\psi}_t \right\| \right\}$

**end for**

**Output:** $(\mathbf{u}_T, \mathbf{v}_T) \to (\mathbf{u}^*, \mathbf{v}^*)$ as $T \to \infty$. $\hspace{3cm} \{ (\phi_T, \psi_T) \to (\phi, \psi) \}$

---

It is straightforward to check that the nonzero eigenvalues of $\mathbf{C}$ are:

$$\rho_1 \geq \cdots \geq \rho_r \geq -\rho_r \geq \cdots \geq -\rho_1,$$

with corresponding eigenvectors $\frac{1}{\sqrt{2}} \begin{bmatrix} \mathbf{a}_1 \\ \mathbf{b}_1 \end{bmatrix}, \ldots, \frac{1}{\sqrt{2}} \begin{bmatrix} \mathbf{a}_r \\ \mathbf{b}_r \end{bmatrix}, \frac{1}{\sqrt{2}} \begin{bmatrix} \mathbf{a}_r \\ -\mathbf{b}_r \end{bmatrix}, \ldots, \frac{1}{\sqrt{2}} \begin{bmatrix} \mathbf{a}_1 \\ -\mathbf{b}_1 \end{bmatrix}.$

The key observation is that Algorithm 1 effectively runs a variant of power iterations on $\mathbf{C}$ to extract its top eigenvector. To see this, make the following change of variables

$$\phi_t = \mathbf{\Sigma}_{xx}^{\frac{1}{2}} \mathbf{u}_t, \qquad \psi_t = \mathbf{\Sigma}_{yy}^{\frac{1}{2}} \mathbf{v}_t, \qquad \tilde{\phi}_t = \mathbf{\Sigma}_{xx}^{\frac{1}{2}} \tilde{\mathbf{u}}_t, \qquad \tilde{\psi}_t = \mathbf{\Sigma}_{yy}^{\frac{1}{2}} \tilde{\mathbf{v}}_t. \tag{4}$$

Then we can equivalently rewrite the steps of Algorithm 1 in the new variables as in $\{\}$ of each line.

Observe that the iterates are updated as follows from step $t-1$ to step $t$:

$$\begin{bmatrix} \tilde{\phi}_t \\ \tilde{\psi}_t \end{bmatrix} \leftarrow \begin{bmatrix} \mathbf{0} & \mathbf{T} \\ \mathbf{T}^\top & \mathbf{0} \end{bmatrix} \begin{bmatrix} \phi_{t-1} \\ \psi_{t-1} \end{bmatrix}, \qquad \begin{bmatrix} \phi_t \\ \psi_t \end{bmatrix} \leftarrow \begin{bmatrix} \tilde{\phi}_t / \| \tilde{\phi}_t \| \\ \tilde{\psi}_t / \| \tilde{\psi}_t \| \end{bmatrix}. \tag{5}$$

Except for the special normalization steps which rescale the two sets of variables separately, Algorithm 1 is very similar to the power iterations [8].

We show the convergence rate of ALS below (see its proof in Appendix A). The first measure of progress is the alignment of $\phi_t$ to $\phi$ and the alignment of $\psi_t$ to $\psi$, i.e., $(\phi_t^\top \phi)^2 = (\mathbf{u}_t^\top \mathbf{\Sigma}_{xx} \mathbf{u}^*)^2$ and $(\psi_t^\top \psi)^2 = (\mathbf{v}_t^\top \mathbf{\Sigma}_{yy} \mathbf{v}^*)^2$. The maximum value for such alignments is 1, achieved when the iterates completely align with the optimal solution. The second natural measure of progress is the objective of (1), i.e., $\mathbf{u}_t^\top \mathbf{\Sigma}_{xy} \mathbf{v}_t$, with the maximum value being $\rho_1$.

***Theorem*** 1 (Convergence of Algorithm 1). Let $\mu := \min \left( (\mathbf{u}_0^\top \mathbf{\Sigma}_{xx} \mathbf{u}^*)^2, (\mathbf{v}_0^\top \mathbf{\Sigma}_{yy} \mathbf{v}^*)^2 \right) > 0$.[4] Then for $t \geq \lceil \frac{\rho_1^2}{\rho_1^2 - \rho_2^2} \log \left( \frac{1}{\mu \eta} \right) \rceil$, we have in Algorithm 1 that $\min \left( (\mathbf{u}_t^\top \mathbf{\Sigma}_{xx} \mathbf{u}^*)^2, (\mathbf{v}_t^\top \mathbf{\Sigma}_{yy} \mathbf{v}^*)^2 \right) \geq 1 - \eta$, and $\mathbf{u}_t^\top \mathbf{\Sigma}_{xy} \mathbf{v}_t \geq \rho_1 (1 - 2\eta)$.

**Remarks** We have assumed a nonzero singular value gap in Theorem 1 to obtain linear convergence in both the alignments and the objective. When there exists no singular value gap, the top singular vector pair is not unique and it is no longer meaningful to measure the alignments. Nonetheless, it is possible to extend our proof to obtain sublinear convergence for the objective in this case.

Observe that, besides the steps of normalization to unit length, the basic operation in each iteration of Algorithm 1 is of the form $\tilde{\mathbf{u}}_t \leftarrow \mathbf{\Sigma}_{xx}^{-1} \mathbf{\Sigma}_{xy} \mathbf{v}_{t-1} = (\frac{1}{N} \mathbf{X} \mathbf{X}^\top + \gamma_x \mathbf{I})^{-1} \frac{1}{N} \mathbf{X} \mathbf{Y}^\top \mathbf{v}_{t-1}$, which is equivalent to solving the following regularized least squares (ridge regression) problem

$$\min_{\mathbf{u}} \frac{1}{2N} \left\| \mathbf{u}^\top \mathbf{X} - \mathbf{v}_{t-1}^\top \mathbf{Y} \right\|^2 + \frac{\gamma_x}{2} \left\| \mathbf{u} \right\|^2 \equiv \min_{\mathbf{u}} \frac{1}{N} \sum_{i=1}^{N} \frac{1}{2} \left| \mathbf{u}^\top \mathbf{x}_i - \mathbf{v}_{t-1}^\top \mathbf{y}_i \right|^2 + \frac{\gamma_x}{2} \left\| \mathbf{u} \right\|^2. \tag{6}$$

In the next section, we show that, to maintain the convergence of ALS, it is unnecessary to solve the least squares problems exactly. This enables us to use state-of-the-art SGD methods for solving (6) to sufficient accuracy, and to obtain a globally convergent stochastic algorithm for CCA.

**Algorithm 2** The alternating least squares (ALS) meta-algorithm for CCA.

---

**Input:** Data matrices $\mathbf{X} \in \mathbb{R}^{d_x \times N}$, $\mathbf{Y} \in \mathbb{R}^{d_y \times N}$, regularization parameters $(\gamma_x, \gamma_y)$.

Initialize $\tilde{\mathbf{u}}_0 \in \mathbb{R}^{d_x}$, $\tilde{\mathbf{v}}_0 \in \mathbb{R}^{d_y}$.

$\tilde{\mathbf{u}}_0 \leftarrow \tilde{\mathbf{u}}_0/\sqrt{\tilde{\mathbf{u}}_0^\top \boldsymbol{\Sigma}_{xx} \tilde{\mathbf{u}}_0}, \qquad \tilde{\mathbf{v}}_0 \leftarrow \tilde{\mathbf{v}}_0/\sqrt{\tilde{\mathbf{v}}_0^\top \boldsymbol{\Sigma}_{yy} \tilde{\mathbf{v}}_0}, \qquad \mathbf{u}_0 \leftarrow \tilde{\mathbf{u}}_0, \qquad \mathbf{v}_0 \leftarrow \tilde{\mathbf{v}}_0$

**for** $t = 1, 2, \ldots, T$ **do**

Solve $\min_{\mathbf{u}} f_t(\mathbf{u}) := \dfrac{1}{2N}\left\|\mathbf{u}^\top \mathbf{X} - \mathbf{v}_{t-1}^\top \mathbf{Y}\right\|^2 + \dfrac{\gamma_x}{2}\left\|\mathbf{u}\right\|^2$ with initialization $\tilde{\mathbf{u}}_{t-1}$, and output approximate solution $\tilde{\mathbf{u}}_t$ satisfying $f_t(\tilde{\mathbf{u}}_t) \le \min_{\mathbf{u}} f_t(\mathbf{u}) + \epsilon$.

Solve $\min_{\mathbf{v}} g_t(\mathbf{v}) := \dfrac{1}{2N}\left\|\mathbf{v}^\top \mathbf{Y} - \mathbf{u}_{t-1}^\top \mathbf{X}\right\|^2 + \dfrac{\gamma_y}{2}\left\|\mathbf{v}\right\|^2$ with initialization $\tilde{\mathbf{v}}_{t-1}$, and output approximate solution $\tilde{\mathbf{v}}_t$ satisfying $g_t(\tilde{\mathbf{v}}_t) \le \min_{\mathbf{v}} g_t(\mathbf{v}) + \epsilon$.

$\mathbf{u}_t \leftarrow \tilde{\mathbf{u}}_t/\sqrt{\tilde{\mathbf{u}}_t^\top \boldsymbol{\Sigma}_{xx} \tilde{\mathbf{u}}_t}, \qquad \mathbf{v}_t \leftarrow \tilde{\mathbf{v}}_t/\sqrt{\tilde{\mathbf{v}}_t^\top \boldsymbol{\Sigma}_{yy} \tilde{\mathbf{v}}_t}$

**end for**

**Output:** $(\mathbf{u}_T, \mathbf{v}_T)$ is the approximate solution to CCA.

---

## 3 Our algorithms

### 3.1 Algorithm I: Alternating least squares (ALS) with variance reduction

Our first algorithm consists of two nested loops. The outer loop runs inexact power iterations while the inner loop uses advanced stochastic optimization methods, e.g., stochastic variance reduced gradient (SVRG, [9]) to obtain approximate matrix-vector multiplications. A sketch of our algorithm is provided in Algorithm 2. We make the following observations from this algorithm.

**Connection to previous work** At step $t$, if we optimize $f_t(\mathbf{u})$ and $g_t(\mathbf{v})$ crudely by a single batch gradient descent step from the initialization $(\tilde{\mathbf{u}}_{t-1}, \tilde{\mathbf{v}}_{t-1})$, we obtain the following update rule:

$$\tilde{\mathbf{u}}_t \leftarrow \tilde{\mathbf{u}}_{t-1} - 2\xi\, \mathbf{X}(\mathbf{X}^\top \tilde{\mathbf{u}}_{t-1} - \mathbf{Y}^\top \mathbf{v}_{t-1})/N, \qquad \mathbf{u}_t \leftarrow \tilde{\mathbf{u}}_t/\sqrt{\tilde{\mathbf{u}}_t^\top \boldsymbol{\Sigma}_{xx} \tilde{\mathbf{u}}_t}$$

$$\tilde{\mathbf{v}}_t \leftarrow \tilde{\mathbf{v}}_{t-1} - 2\xi\, \mathbf{Y}(\mathbf{Y}^\top \tilde{\mathbf{v}}_{t-1} - \mathbf{X}^\top \mathbf{u}_{t-1})/N, \qquad \mathbf{v}_t \leftarrow \tilde{\mathbf{v}}_t/\sqrt{\tilde{\mathbf{v}}_t^\top \boldsymbol{\Sigma}_{yy} \tilde{\mathbf{v}}_t}$$

where $\xi > 0$ is the stepsize (assuming $\gamma_x = \gamma_y = 0$). This coincides with the `AppGrad` algorithm of [3, Algorithm 3], for which only local convergence is shown. Since the objectives $f_t(\mathbf{u})$ and $g_t(\mathbf{v})$ decouple over training samples, it is convenient to apply SGD methods to them. This observation motivated the stochastic CCA algorithms of [3, 4]. We note however, no global convergence guarantee was shown for these stochastic CCA algorithms, and the key to our convergent algorithm is to solve the least squares problems to *sufficient* accuracy.

**Warm-start** Observe that for different $t$, the least squares problems $f_t(\mathbf{u})$ only differ in their targets as $\mathbf{v}_t$ changes over time. Since $\mathbf{v}_{t-1}$ is close to $\mathbf{v}_t$ (especially when near convergence), we may use $\tilde{\mathbf{u}}_t$ as initialization for minimizing $f_{t+1}(\mathbf{u})$ with an iterative algorithm.

**Normalization** At the end of each outer loop, Algorithm 2 implements exact normalization of the form $\mathbf{u}_t \leftarrow \tilde{\mathbf{u}}_t/\sqrt{\tilde{\mathbf{u}}_t^\top \boldsymbol{\Sigma}_{xx} \tilde{\mathbf{u}}_t}$ to ensure the constraints, where $\tilde{\mathbf{u}}_t^\top \boldsymbol{\Sigma}_{xx} \tilde{\mathbf{u}}_t = \frac{1}{N}(\tilde{\mathbf{u}}_t^\top \mathbf{X})(\tilde{\mathbf{u}}_t^\top \mathbf{X})^\top + \gamma_x \|\tilde{\mathbf{u}}_t\|^2$ requires computing the projection of the training set $\tilde{\mathbf{u}}_t^\top \mathbf{X}$. However, this does not introduce extra computation because we also compute this projection for the batch gradient used by SVRG (at the beginning of time step $t+1$). In contrast, the stochastic algorithms of [3, 4] (possibly adaptively) estimate the covariance matrix from a minibatch of training samples and use the estimated covariance for normalization. This is because their algorithms perform normalizations after each update and thus need to avoid computing the projection of the entire training set frequently. But as a result, their inexact normalization steps introduce noise to the algorithms.

**Input sparsity** For high dimensional sparse data (such as those used in natural language processing [10]), an advantage of gradient based methods over the closed-form solution is that the former takes into account the input sparsity. For sparse inputs, the time complexity of our algorithm depends on $nnz(\mathbf{X}, \mathbf{Y})$, i.e., the total number of nonzeros in the inputs instead of $dN$.

**Canonical ridge** When $(\gamma_x, \gamma_y) > 0$, $f_t(\mathbf{u})$ and $g_t(\mathbf{v})$ are guaranteed to be strongly convex due to the $\ell_2$ regularizations, in which case SVRG converges linearly. It is therefore beneficial to use

small nonzero regularization for improved computational efficiency, especially for high dimensional datasets where inputs $\mathbf{X}$ and $\mathbf{Y}$ are approximately low-rank.

**Convergence** By the analysis of inexact power iterations where the least squares problems are solved (or the matrix-vector multiplications are computed) only up to necessary accuracy, we provide the following theorem for the convergence of Algorithm 2 (see its proof in Appendix B). The key to our analysis is to bound the distances between the iterates of Algorithm 2 and that of Algorithm 1 at all time steps, and when the errors of the least squares problems are sufficiently small (at the level of $\eta^2$), the iterates of the two algorithms have the same quality.

***Theorem*** 2 (Convergence of Algorithm 2). Fix $T \geq \lceil \frac{\rho_1^2}{\rho_1^2 - \rho_2^2} \log\left(\frac{2}{\mu\eta}\right) \rceil$, and set $\epsilon(T) \leq \frac{\eta^2 \rho_r^2}{128} \cdot \left(\frac{(2\rho_1/\rho_r)-1}{(2\rho_1/\rho_r)^T-1}\right)^2$ in Algorithm 2. Then we have $\mathbf{u}_T^\top \boldsymbol{\Sigma}_{xx} \mathbf{u}_T = \mathbf{v}_T^\top \boldsymbol{\Sigma}_{yy} \mathbf{v}_T = 1$, $\min\left((\mathbf{u}_T^\top \boldsymbol{\Sigma}_{xx} \mathbf{u}^*)^2, (\mathbf{v}_T^\top \boldsymbol{\Sigma}_{yy} \mathbf{v}^*)^2\right) \geq 1 - \eta$, and $\mathbf{u}_T^\top \boldsymbol{\Sigma}_{xy} \mathbf{v}_T \geq \rho_1(1 - 2\eta)$.

### 3.1.1 Stochastic optimization of regularized least squares

We now discuss the inner loop of Algorithm 2, which approximately solves problems of the form (6). Owing to the finite-sum structure of (6), several stochastic optimization methods such as SAG [11], SDCA [12] and SVRG [9], provide linear convergence rates. All these algorithms can be readily applied to (6); we choose SVRG since it is memory efficient and easy to implement. We also apply the recently developed accelerations techniques for first order optimization methods [13, 14] to obtain an accelerated SVRG (ASVRG) algorithm. We give the sketch of SVRG for (6) in Appendix C.

Note that $f(\mathbf{u}) = \frac{1}{N}\sum_{i=1}^{N} f^i(\mathbf{u})$ where each component $f^i(\mathbf{u}) = \frac{1}{2}\left|\mathbf{u}^\top \mathbf{x}_i - \mathbf{v}^\top \mathbf{y}_i\right|^2 + \frac{\gamma_x}{2}\|\mathbf{u}\|^2$ is $\|\mathbf{x}_i\|^2$-smooth, and $f(\mathbf{u})$ is $\sigma_{\min}(\boldsymbol{\Sigma}_{xx})$-strongly convex[5] with $\sigma_{\min}(\boldsymbol{\Sigma}_{xx}) \geq \gamma_x$. We show in Appendix D that the initial suboptimality for minimizing $f_t(\mathbf{u})$ is upper-bounded by constant when using the warm-starts. We quote the convergence rates of SVRG [9] and ASVRG [14] below.

***Lemma*** 3. The SVRG algorithm [9] finds a vector $\tilde{\mathbf{u}}$ satisfying[6] $\mathbb{E}[f(\tilde{\mathbf{u}})] - \min_{\mathbf{u}} f(\mathbf{u}) \leq \epsilon$ in time $\mathcal{O}\left(d_x\left(N + \kappa_x\right)\log\left(\frac{1}{\epsilon}\right)\right)$ where $\kappa_x = \frac{\max_i\|\mathbf{x}_i\|^2}{\sigma_{\min}(\boldsymbol{\Sigma}_{xx})}$. The ASVRG algorithm [14] finds a such solution in time $\mathcal{O}\left(d_x\sqrt{N\kappa_x}\log\left(\frac{1}{\epsilon}\right)\right)$.

**Remarks** As mentioned in [14], the acceleration version provides speedup over normal SVRG only when $\kappa_x > N$ and we only show the dominant term in the above complexity.

By combining the iteration complexity of the outer loop (Theorem 2) and the time complexity of the inner loop (Lemma 3), we obtain the total time complexity of $\tilde{\mathcal{O}}\left(d\left(N + \kappa\right)\left(\frac{\rho_1^2}{\rho_1^2 - \rho_2^2}\right)^2 \cdot \log^2\left(\frac{1}{\eta}\right)\right)$ for ALS+SVRG and $\tilde{\mathcal{O}}\left(d\sqrt{N\kappa}\left(\frac{\rho_1^2}{\rho_1^2 - \rho_2^2}\right)^2 \cdot \log^2\left(\frac{1}{\eta}\right)\right)$ for ALS+ASVRG, where $\kappa := \max\left(\frac{\max_i\|\mathbf{x}_i\|^2}{\sigma_{\min}(\boldsymbol{\Sigma}_{xx})}, \frac{\max_i\|\mathbf{y}_i\|^2}{\sigma_{\min}(\boldsymbol{\Sigma}_{yy})}\right)$ and $\tilde{\mathcal{O}}(\cdot)$ hides poly-logarithmic dependences on $\frac{1}{\mu}$ and $\frac{1}{\rho_r}$. Our algorithm does not require the initialization to be close to the optimum and converges globally. For comparison, the locally convergent `AppGrad` has a time complexity [3, Theorem 2.1] of $\tilde{\mathcal{O}}\left(dN\kappa'\frac{\rho_1^2}{\rho_1^2 - \rho_2^2} \cdot \log\left(\frac{1}{\eta}\right)\right)$, where $\kappa' := \max\left(\frac{\sigma_{\max}(\boldsymbol{\Sigma}_{xx})}{\sigma_{\min}(\boldsymbol{\Sigma}_{xx})}, \frac{\sigma_{\max}(\boldsymbol{\Sigma}_{yy})}{\sigma_{\min}(\boldsymbol{\Sigma}_{yy})}\right)$. Note, in this complexity, the dataset size $N$ and the least squares condition number $\kappa'$ are multiplied together because `AppGrad` essentially uses batch gradient descent as the least squares solver. Within our framework, we can use accelerated gradient descent (AGD, [15]) instead and obtain a globally convergent algorithm with a total time complexity of $\tilde{\mathcal{O}}\left(dN\sqrt{\kappa'}\left(\frac{\rho_1^2}{\rho_1^2 - \rho_2^2}\right)^2 \cdot \log^2\left(\frac{1}{\eta}\right)\right)$.

### 3.2 Algorithm II: Shift-and-invert preconditioning (SI) with variance reduction

The second algorithm is inspired by the shift-and-invert preconditioning method for PCA [16, 17]. Instead of running power iterations on $\mathbf{C}$ as defined in (3), we will be running power iterations on

$$\mathbf{M}_\lambda = (\lambda\mathbf{I} - \mathbf{C})^{-1} = \begin{bmatrix} \lambda\mathbf{I} & -\mathbf{T} \\ -\mathbf{T}^\top & \lambda\mathbf{I} \end{bmatrix}^{-1} \in \mathbb{R}^{d\times d}, \tag{7}$$

where $\lambda > \rho_1$. It is straightforward to check that $\mathbf{M}_\lambda$ is positive definite and its eigenvalues are:

$$\frac{1}{\lambda - \rho_1} \geq \cdots \geq \frac{1}{\lambda - \rho_r} \geq \cdots \geq \frac{1}{\lambda + \rho_r} \geq \cdots \geq \frac{1}{\lambda + \rho_1},$$

with eigenvectors $\frac{1}{\sqrt{2}} \begin{bmatrix} \mathbf{a}_1 \\ \mathbf{b}_1 \end{bmatrix}, \ldots, \frac{1}{\sqrt{2}} \begin{bmatrix} \mathbf{a}_r \\ \mathbf{b}_r \end{bmatrix}, \ldots, \frac{1}{\sqrt{2}} \begin{bmatrix} \mathbf{a}_r \\ -\mathbf{b}_r \end{bmatrix}, \ldots, \frac{1}{\sqrt{2}} \begin{bmatrix} \mathbf{a}_1 \\ -\mathbf{b}_1 \end{bmatrix}$.

The main idea behind shift-and-invert power iterations is that when $\lambda - \rho_1 = c(\rho_1 - \rho_2)$ with $c \sim \mathcal{O}(1)$, the relative eigenvalue gap of $\mathbf{M}_\lambda$ is large and so power iterations on $\mathbf{M}_\lambda$ converges quickly. Our shift-and-invert preconditioning (SI) meta-algorithm for CCA is sketched in Algorithm 3 (in Appendix E due to space limit) and it proceeds in two phases.

### 3.2.1 Phase I: shift-and-invert preconditioning for eigenvectors of $\mathbf{M}_\lambda$

Using an estimate of the singular value gap $\tilde{\Delta}$ and starting from an over-estimate of $\rho_1$ ($1 + \tilde{\Delta}$ suffices), the algorithm gradually shrinks $\lambda_{(s)}$ towards $\rho_1$ by crudely estimating the leading eigenvector/eigenvalues of each $\mathbf{M}_{\lambda_{(s)}}$ along the way and shrinking the gap $\lambda_{(s)} - \rho_1$, until we reach a $\lambda_{(f)} \in (\rho_1, \rho_1 + c(\rho_1 - \rho_2))$ where $c \sim \mathcal{O}(1)$. Afterwards, the algorithm fixes $\lambda_{(f)}$ and runs inexact power iterations on $\mathbf{M}_{\lambda_{(f)}}$ to obtain an accurate estimate of its leading eigenvector. Note in this phase, power iterations implicitly operate on the concatenated variables $\frac{1}{\sqrt{2}} \begin{bmatrix} \mathbf{\Sigma}_{xx}^{\frac{1}{2}} \tilde{\mathbf{u}}_t \\ \mathbf{\Sigma}_{yy}^{\frac{1}{2}} \tilde{\mathbf{v}}_t \end{bmatrix}$ and

$\frac{1}{\sqrt{2}} \begin{bmatrix} \mathbf{\Sigma}_{xx}^{\frac{1}{2}} \mathbf{u}_t \\ \mathbf{\Sigma}_{yy}^{\frac{1}{2}} \mathbf{v}_t \end{bmatrix}$ in $\mathbb{R}^d$ (but without ever computing $\mathbf{\Sigma}_{xx}^{\frac{1}{2}}$ and $\mathbf{\Sigma}_{yy}^{\frac{1}{2}}$).

**Matrix-vector multiplication**   The matrix-vector multiplications in Phase I have the form

$$\begin{bmatrix} \tilde{\mathbf{u}}_t \\ \tilde{\mathbf{v}}_t \end{bmatrix} \leftarrow \begin{bmatrix} \lambda\mathbf{\Sigma}_{xx} & -\mathbf{\Sigma}_{xy} \\ -\mathbf{\Sigma}_{xy}^\top & \lambda\mathbf{\Sigma}_{yy} \end{bmatrix}^{-1} \begin{bmatrix} \mathbf{\Sigma}_{xx} & \\ & \mathbf{\Sigma}_{yy} \end{bmatrix} \begin{bmatrix} \mathbf{u}_{t-1} \\ \mathbf{v}_{t-1} \end{bmatrix}, \tag{8}$$

where $\lambda$ varies over time in order to locate $\lambda_{(f)}$. This is equivalent to solving

$$\begin{bmatrix} \tilde{\mathbf{u}}_t \\ \tilde{\mathbf{v}}_t \end{bmatrix} \leftarrow \min_{\mathbf{u},\mathbf{v}} \frac{1}{2} \begin{bmatrix} \mathbf{u}^\top \mathbf{v}^\top \end{bmatrix} \begin{bmatrix} \lambda\mathbf{\Sigma}_{xx} & -\mathbf{\Sigma}_{xy} \\ -\mathbf{\Sigma}_{xy}^\top & \lambda\mathbf{\Sigma}_{yy} \end{bmatrix} \begin{bmatrix} \mathbf{u} \\ \mathbf{v} \end{bmatrix} - \mathbf{u}^\top \mathbf{\Sigma}_{xx} \mathbf{u}_{t-1} - \mathbf{v}^\top \mathbf{\Sigma}_{yy} \mathbf{v}_{t-1}.$$

And as in ALS, this least squares problem can be further written as finite-sum:

$$\min_{\mathbf{u},\mathbf{v}} \quad h_t(\mathbf{u}, \mathbf{v}) = \frac{1}{N} \sum_{i=1}^{N} h_t^i(\mathbf{u}, \mathbf{v}) \qquad \text{where} \tag{9}$$

$$h_t^i(\mathbf{u}, \mathbf{v}) = \frac{1}{2} \begin{bmatrix} \mathbf{u}^\top \mathbf{v}^\top \end{bmatrix} \begin{bmatrix} \lambda\left(\mathbf{x}_i\mathbf{x}_i^\top + \gamma_x\mathbf{I}\right) & -\mathbf{x}_i\mathbf{y}_i^\top \\ -\mathbf{y}_i\mathbf{x}_i^\top & \lambda\left(\mathbf{y}_i\mathbf{y}_i^\top + \gamma_y\mathbf{I}\right) \end{bmatrix} \begin{bmatrix} \mathbf{u} \\ \mathbf{v} \end{bmatrix} - \mathbf{u}^\top \mathbf{\Sigma}_{xx} \mathbf{u}_{t-1} - \mathbf{v}^\top \mathbf{\Sigma}_{yy} \mathbf{v}_{t-1}.$$

We could directly apply SGD methods to this problem as before.

**Normalization**   The normalization steps in Phase I have the form

$$\begin{bmatrix} \mathbf{u}_t \\ \mathbf{v}_t \end{bmatrix} \leftarrow \sqrt{2} \begin{bmatrix} \tilde{\mathbf{u}}_t \\ \tilde{\mathbf{v}}_t \end{bmatrix} \Big/ \sqrt{\tilde{\mathbf{u}}_t^\top \mathbf{\Sigma}_{xx} \tilde{\mathbf{u}}_t + \tilde{\mathbf{v}}_t^\top \mathbf{\Sigma}_{yy} \tilde{\mathbf{v}}_t},$$

and so the following remains true for the normalized iterates in Phase I:

$$\mathbf{u}_t^\top \mathbf{\Sigma}_{xx} \mathbf{u}_t + \mathbf{v}_t^\top \mathbf{\Sigma}_{yy} \mathbf{v}_t = 2, \qquad \text{for} \quad t = 1, \ldots, T. \tag{10}$$

Unlike the normalizations in ALS, the iterates $\mathbf{u}_t$ and $\mathbf{v}_t$ in Phase I do *not* satisfy the original CCA constraints, and this is taken care of in Phase II.

We have the following convergence guarantee for Phase I (see its proof in Appendix F).

***Theorem*** 4 (Convergence of Algorithm 3, Phase I). Let $\Delta = \rho_1 - \rho_2 \in (0, 1]$, and $\tilde{\mu} := \frac{1}{4}\left(\mathbf{u}_0^\top \mathbf{\Sigma}_{xx} \mathbf{u}^* + \mathbf{v}_0^\top \mathbf{\Sigma}_{yy} \mathbf{v}^*\right)^2 > 0$, and $\tilde{\Delta} \in [c_1\Delta, c_2\Delta]$ where $0 < c_1 \leq c_2 \leq 1$. Set $m_1 = \lceil 8 \log\left(\frac{16}{\tilde{\mu}}\right)\rceil$, $m_2 = \lceil \frac{5}{4} \log\left(\frac{128}{\tilde{\mu}\eta^2}\right)\rceil$, and $\tilde{\epsilon} \leq \min\left(\frac{1}{3084}\left(\frac{\tilde{\Delta}}{18}\right)^{m_1-1}, \frac{\eta^4}{4^{10}}\left(\frac{\tilde{\Delta}}{18}\right)^{m_2-1}\right)$ in Algorithm 3. Then the $(\mathbf{u}_T, \mathbf{v}_T)$ output by Phase I of Algorithm 3 satisfies (10) and

$$\frac{1}{4}(\mathbf{u}_T^\top \mathbf{\Sigma}_{xx} \mathbf{u}^* + \mathbf{v}_T^\top \mathbf{\Sigma}_{yy} \mathbf{v}^*)^2 \geq 1 - \frac{\eta^2}{64}, \tag{11}$$

and the number of calls to the least squares solver of $h_t(\mathbf{u}, \mathbf{v})$ is $\mathcal{O}\left(\log\left(\frac{1}{\tilde{\mu}}\right)\log\left(\frac{1}{\Delta}\right) + \log\left(\frac{1}{\tilde{\mu}\eta^2}\right)\right)$.

### 3.2.2 Phase II: final normalization

In order to satisfy the CCA constraints, we perform a last normalization

$$\hat{\mathbf{u}} \leftarrow \mathbf{u}_T / \sqrt{\mathbf{u}_T^\top \mathbf{\Sigma}_{xx} \mathbf{u}_T}, \qquad \hat{\mathbf{v}} \leftarrow \mathbf{v}_T / \sqrt{\mathbf{v}_T^\top \mathbf{\Sigma}_{yy} \mathbf{v}_T}. \tag{12}$$

And we output $(\hat{\mathbf{u}}, \hat{\mathbf{v}})$ as our final approximate solution to (1). We show that this step does not cause much loss in the alignments, as stated below (see it proof in Appendix G).

***Theorem*** 5 (Convergence of Algorithm 3, Phase II). *Let Phase I of Algorithm 3 outputs $(\mathbf{u}_T, \mathbf{v}_T)$ that satisfy (11). Then after (12), we obtain an approximate solution $(\hat{\mathbf{u}}, \hat{\mathbf{v}})$ to (1) such that $\hat{\mathbf{u}}^\top \mathbf{\Sigma}_{xx} \hat{\mathbf{u}} = \hat{\mathbf{v}}^\top \mathbf{\Sigma}_{yy} \hat{\mathbf{v}} = 1, \min\left( (\hat{\mathbf{u}}^\top \mathbf{\Sigma}_{xx} \mathbf{u}^*)^2, (\hat{\mathbf{v}}^\top \mathbf{\Sigma}_{yy} \mathbf{v}^*)^2 \right) \geq 1 - \eta$, and $\hat{\mathbf{u}}^\top \mathbf{\Sigma}_{xy} \hat{\mathbf{v}} \geq \rho_1 (1 - 2\eta)$.*

### 3.2.3 Time complexity

We have shown in Theorem 4 that Phase I only approximately solves a small number of instances of (9). The normalization steps (10) require computing the projections of the traning set which are reused for computing batch gradients of (9). The final normalization (12) is done only once and costs $\mathcal{O}(dN)$. Therefore, the time complexity of our algorithm mainly comes from solving the least squares problems (9) using SGD methods in a blackbox fashion. And the time complexity for SGD methods depends on the condition number of (9). Denote

$$\mathbf{Q}_\lambda = \begin{bmatrix} \lambda \mathbf{\Sigma}_{xx} & -\mathbf{\Sigma}_{xy} \\ -\mathbf{\Sigma}_{xy}^\top & \lambda \mathbf{\Sigma}_{yy} \end{bmatrix} = \begin{bmatrix} \mathbf{\Sigma}_{xx}^{\frac{1}{2}} & \\ & \mathbf{\Sigma}_{yy}^{\frac{1}{2}} \end{bmatrix} \begin{bmatrix} \lambda \mathbf{I} & -\mathbf{T} \\ -\mathbf{T}^\top & \lambda \mathbf{I} \end{bmatrix} \begin{bmatrix} \mathbf{\Sigma}_{xx}^{\frac{1}{2}} & \\ & \mathbf{\Sigma}_{yy}^{\frac{1}{2}} \end{bmatrix}. \tag{13}$$

It is clear that
$$\sigma_{\max}(\mathbf{Q}_\lambda) \leq (\lambda + \rho_1) \cdot \max\left( \sigma_{\max}(\mathbf{\Sigma}_{xx}), \sigma_{\max}(\mathbf{\Sigma}_{yy}) \right),$$
$$\sigma_{\min}(\mathbf{Q}_\lambda) \geq (\lambda - \rho_1) \cdot \min\left( \sigma_{\min}(\mathbf{\Sigma}_{xx}), \sigma_{\min}(\mathbf{\Sigma}_{yy}) \right).$$

We have shown in the proof of Theorem 4 that $\frac{\lambda + \rho_1}{\lambda - \rho_1} \leq \frac{9}{\Delta} \leq \frac{9}{c_1 \Delta}$ throughout Algorithm 3 (cf. Lemma 10, Appendix F.2), and thus the condtion number for AGD is $\frac{\sigma_{\max}(\mathbf{Q}_\lambda)}{\sigma_{\min}(\mathbf{Q}_\lambda)} \leq \frac{9/c_1}{\rho_1 - \rho_2} \tilde{\kappa}'$, where $\tilde{\kappa}' := \frac{\max(\sigma_{\max}(\mathbf{\Sigma}_{xx}), \sigma_{\max}(\mathbf{\Sigma}_{yy}))}{\min(\sigma_{\min}(\mathbf{\Sigma}_{xx}), \sigma_{\min}(\mathbf{\Sigma}_{yy}))}$. For SVRG/ASVRG, the relevant condition number depends on the gradient Lipschitz constant of individual components. We show in Appendix H (Lemma 12) that the relevant condition number is at most $\frac{9/c_1}{\rho_1 - \rho_2} \tilde{\kappa}$, where $\tilde{\kappa} := \frac{\max_i \max(\|\mathbf{x}_i\|^2, \|\mathbf{y}_i\|^2)}{\min(\sigma_{\min}(\mathbf{\Sigma}_{xx}), \sigma_{\min}(\mathbf{\Sigma}_{yy}))}$. An interesting issue for SVRG/ASVRG is that, depending on the value of $\lambda$, the independent components $h_t^i(\mathbf{u}, \mathbf{v})$ may be nonconvex. If $\lambda \geq 1$, each component is still guaranteed to by convex; otherwise, some components might be non-convex, with the overall average $\frac{1}{N} \sum_{i=1}^N \mathbf{h}_t^i$ being convex. In the later case, we use the modified analysis of SVRG [16, Appendix B] for its time complexity. We use warm-start in SI as in ALS, and the initial suboptimality for each subproblem can be bounded similarly.

The total time complexities of our SI meta-algorithm are given in Table 1. Note that $\tilde{\kappa}$ (or $\tilde{\kappa}'$) and $\frac{1}{\rho_1 - \rho_2}$ are multiplied together, giving the effective condition number. When using SVRG as the least squares solver, we obtain the total time complexity of $\tilde{\mathcal{O}}\left( d(N + \tilde{\kappa} \frac{1}{\rho_1 - \rho_2}) \cdot \log^2\left( \frac{1}{\eta} \right) \right)$ if all components are convex, and $\tilde{\mathcal{O}}\left( d(N + (\tilde{\kappa} \frac{1}{\rho_1 - \rho_2})^2) \cdot \log^2\left( \frac{1}{\eta} \right) \right)$ otherwise. When using ASVRG, we have $\tilde{\mathcal{O}}\left( d\sqrt{N} \sqrt{\tilde{\kappa}} \sqrt{\frac{1}{\rho_1 - \rho_2}} \cdot \log^2\left( \frac{1}{\eta} \right) \right)$ if all components are convex, and $\tilde{\mathcal{O}}\left( dN^{\frac{3}{4}} \sqrt{\tilde{\kappa}} \sqrt{\frac{1}{\rho_1 - \rho_2}} \cdot \log^2\left( \frac{1}{\eta} \right) \right)$ otherwise. Here $\tilde{\mathcal{O}}(\cdot)$ hides poly-logarithmic dependences on $\frac{1}{\mu}$ and $\frac{1}{\Delta}$. It is remarkable that the SI meta-algorithm is able to separate the dependence of dataset size $N$ from other parameters in the time complexities.

**Parallel work** In a parallel work [6], the authors independently proposed a similar ALS algorithm[7], and they solve the least squares problems using AGD. The time complexity of their algorithm for extracting the first canonical correlation is $\tilde{\mathcal{O}}\left( dN \sqrt{\kappa'} \frac{\rho_1^2}{\rho_1^2 - \rho_2^2} \cdot \log\left( \frac{1}{\eta} \right) \right)$, which has linear dependence on $\frac{\rho_1^2}{\rho_1^2 - \rho_2^2} \log\left( \frac{1}{\eta} \right)$ (so their algorithm is linearly convergent, but our complexity for ALS+AGD has quadratic dependence on this factor), but typically worse dependence on $N$ and $\kappa'$ (see remarks in Section 3.1.1). Moreover, our SI algorithm tends to significantly outperform ALS theoretically and empirically. It is future work to remove extra $\log\left( \frac{1}{\eta} \right)$ dependence in our analysis.

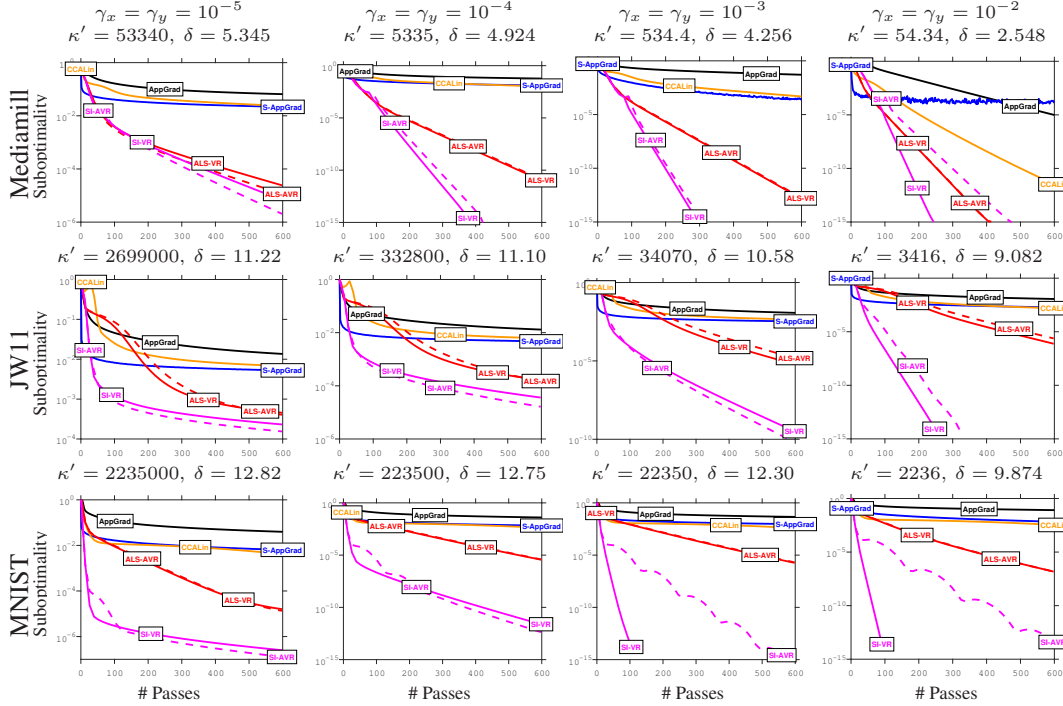

Figure 1: Comparison of suboptimality vs. # passes for different algorithms. For each dataset and regularization parameters $(\gamma_x, \gamma_y)$, we give $\kappa' = \max\left(\frac{\sigma_{\max}(\boldsymbol{\Sigma}_{xx})}{\sigma_{\min}(\boldsymbol{\Sigma}_{xx})}, \frac{\sigma_{\max}(\boldsymbol{\Sigma}_{yy})}{\sigma_{\min}(\boldsymbol{\Sigma}_{yy})}\right)$ and $\delta = \frac{\rho_1^2}{\rho_1^2 - \rho_2^2}$.

**Extension to multi-dimensional projections** To extend our algorithms to $L$-dimensional projections, we can extract the dimensions sequentially and remove the explained correlation from $\boldsymbol{\Sigma}_{xy}$ each time we extract a new dimension [18]. For the ALS meta-algorithm, a cleaner approach is to extract the $L$ dimensions simultaneously using (inexact) orthogonal iterations [8], in which case the subproblems become multi-dimensional regressions and our normalization steps are of the form $\mathbf{U}_t \leftarrow \tilde{\mathbf{U}}_t(\tilde{\mathbf{U}}_t^\top \boldsymbol{\Sigma}_{xx}\tilde{\mathbf{U}}_t)^{-\frac{1}{2}}$ (the same normalization is used by [3, 4]). Such normalization involves the eigenvalue decomposition of a $L \times L$ matrix and can be solved exactly as we typically look for low dimensional projections. Our analysis for $L = 1$ can be extended to this scenario and the convergence rate of ALS will depend on the gap between $\rho_L$ and $\rho_{L+1}$.

## 4 Experiments

We demonstrate the proposed algorithms, namely `ALS-VR`, `ALS-AVR`, `SI-VR`, and `SI-AVR`, abbreviated as "meta-algorithm – least squares solver" (VR for SVRG, and AVR for ASVRG) on three real-world datasets: Mediamill [19] ($N = 3 \times 10^4$), JW11 [20] ($N = 3 \times 10^4$), and MNIST [21] ($N = 6 \times 10^4$). We compare our algorithms with batch `AppGrad` and its stochastic version `s-AppGrad` [3], as well as the `CCALin` algorithm in parallel work [6]. For each algorithm, we compare the canonical correlation estimated by the iterates at different number of passes over the data with that of the exact solution by SVD. For each dataset, we vary the regularization parameters $\gamma_x = \gamma_y$ over $\{10^{-5}, 10^{-4}, 10^{-3}, 10^{-2}\}$ to vary the least squares condition numbers, and larger regularization leads to better conditioning. We plot the suboptimality in objective vs. # passes for each algorithm in Figure 1. Experimental details (e.g. SVRG parameters) are given in Appendix I.

We make the following observations from the results. First, the proposed stochastic algorithms significantly outperform batch gradient based methods `AppGrad`/`CCALin`. This is because the least squares condition numbers for these datasets are large, and SVRG enable us to decouple dependences on the dataset size $N$ and the condition number $\kappa$ in the time complexity. Second, `SI-VR` converges faster than `ALS-VR` as it further decouples the dependence on $N$ and the singular value gap of $\mathbf{T}$. Third, inexact normalizations keep the `s-AppGrad` algorithm from converging to an accurate solution. Finally, ASVRG improves over SVRG when the the condition number is large.

### Acknowledgments
Research partially supported by NSF BIGDATA grant 1546500.

## Footnotes

[3]For the ALS meta-algorithm, its enough to consider a per-view conditioning. And when using AGD as the least squares solver, the time complexities dependends on $\sigma_{\max}(\boldsymbol{\Sigma}_{xx})$ instead, which is less than $\max_i \|x_i\|^2$.

[4]One can show that $\mu$ is bounded away from 0 with high probability using random initialization $(\mathbf{u}_0, \mathbf{v}_0)$.

[5]We omit the regularization in these constants, which are typically very small, to have concise expressions.

[6]The expectation is taken over random sampling of component functions. High probability error bounds can be obtained using the Markov's inequality.

[7]Our arxiv preprint for the ALS meta-algorithm was posted before their paper got accepted by ICML 2016.

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
