[Supplementary Material]

# A   Proof of Theorem 1

*Proof.* It is easy to see that by the end of the first iteration of Algorithm 1, $\tilde{\boldsymbol{\psi}}_1$ and $\boldsymbol{\psi}_1$ lie in the span of $\{\mathbf{b}_i\}_{i=1}^r$, while $\tilde{\boldsymbol{\phi}}_1$ and $\boldsymbol{\phi}_1$ lie in the span of $\{\mathbf{a}_i\}_{i=1}^r$. And therefore they remain in these spaces for all $t \geq 1$.

Let us first focus on $\boldsymbol{\phi}_t$. For $t \geq 2$, we observe that

$$\boldsymbol{\phi}_t = \mathbf{T}\boldsymbol{\psi}_{t-1}/\left\|\tilde{\boldsymbol{\phi}}_t\right\| = \mathbf{T}\mathbf{T}^\top \boldsymbol{\phi}_{t-2}/\left(\left\|\tilde{\boldsymbol{\phi}}_t\right\| \cdot \left\|\tilde{\boldsymbol{\psi}}_{t-1}\right\|\right).$$

Since $\left\|\boldsymbol{\phi}_{t-2}\right\| = \left\|\boldsymbol{\phi}_t\right\| = 1$, it is equivalent to using the following updates:

$$\boldsymbol{\phi}_t \leftarrow \mathbf{T}\mathbf{T}^\top \boldsymbol{\phi}_{t-2}, \qquad \boldsymbol{\phi}_t \leftarrow \boldsymbol{\phi}_t/\left\|\boldsymbol{\phi}_t\right\|.$$

This indicates that, Algorithm 1 runs the standard power iterations on $\mathbf{T}\mathbf{T}^\top$ to generate the $\{\boldsymbol{\phi}_t\}_{t \geq 1}$ sequence for every two steps.

(i) For $t = 2, 4, \ldots$, we have $\boldsymbol{\phi}_t = \frac{(\mathbf{T}\mathbf{T}^\top)^{\frac{t}{2}}\boldsymbol{\phi}_0}{\left\|(\mathbf{T}\mathbf{T}^\top)^{\frac{t}{2}}\boldsymbol{\phi}_0\right\|}$. Let $\mathbf{M} = \mathbf{T}\mathbf{T}^\top$, whose nonzero eigenvalues are $\rho_1^2 \geq \rho_2^2 \geq \cdots \geq \rho_r^2 > 0$, with corresponding eigenvectors $\mathbf{a}_1, \ldots, \mathbf{a}_r$. Then, for $i = 1, \ldots, r$,

$$
\begin{aligned}
(\mathbf{a}_i^\top \boldsymbol{\phi}_t)^2 &= \frac{\left(\mathbf{a}_i^\top \mathbf{M}^{\frac{t}{2}}\boldsymbol{\phi}_0\right)^2}{\left\|\mathbf{M}^{\frac{t}{2}}\boldsymbol{\phi}_0\right\|^2} = \frac{\left(\mathbf{a}_i^\top \mathbf{M}^{\frac{t}{2}}\boldsymbol{\phi}_0\right)^2}{\boldsymbol{\phi}_0^\top \mathbf{M}^t \boldsymbol{\phi}_0} = \frac{\left(\rho_i^t \mathbf{a}_i^\top \boldsymbol{\phi}_0\right)^2}{\sum_{j=1}^r \rho_j^{2t}(\mathbf{a}_j^\top \boldsymbol{\phi}_0)^2} = \frac{\left(\mathbf{a}_i^\top \boldsymbol{\phi}_0\right)^2}{\sum_{j=1}^r \left(\frac{\rho_j^2}{\rho_i^2}\right)^t (\mathbf{a}_j^\top \boldsymbol{\phi}_0)^2} \\
&\leq \frac{\left(\mathbf{a}_i^\top \boldsymbol{\phi}_0\right)^2}{\left(\frac{\rho_1^2}{\rho_i^2}\right)^t (\mathbf{a}_1^\top \boldsymbol{\phi}_0)^2} = \frac{\left(\mathbf{a}_i^\top \boldsymbol{\phi}_0\right)^2}{(\mathbf{a}_1^\top \boldsymbol{\phi}_0)^2}\left(\frac{\rho_i^2}{\rho_1^2}\right)^t = \frac{\left(\mathbf{a}_i^\top \boldsymbol{\phi}_0\right)^2}{(\mathbf{a}_1^\top \boldsymbol{\phi}_0)^2}\left(1 - \frac{\rho_1^2 - \rho_i^2}{\rho_1^2}\right)^t \\
&\leq \frac{\left(\mathbf{a}_i^\top \boldsymbol{\phi}_0\right)^2}{(\mathbf{a}_1^\top \boldsymbol{\phi}_0)^2} \exp\left(-\frac{\rho_1^2 - \rho_i^2}{\rho_1^2}t\right).
\end{aligned}
$$

(ii) For $t = 1, 3, \ldots$, we have $\boldsymbol{\phi}_t = \frac{(\mathbf{T}\mathbf{T}^\top)^{\frac{t-1}{2}}\mathbf{T}\boldsymbol{\psi}_0}{\left\|(\mathbf{T}\mathbf{T}^\top)^{\frac{t-1}{2}}\mathbf{T}\boldsymbol{\psi}_0\right\|}$. Let $\mathbf{N} = \mathbf{T}^\top\mathbf{T}$, whose nonzero eigenvalues are $\rho_1^2 \geq \rho_2^2 \geq \cdots \geq \rho_r^2 > 0$, with corresponding eigenvectors $\mathbf{b}_1, \ldots, \mathbf{b}_r$. Then, for $i = 1, \ldots, r$,

$$
\begin{aligned}
(\mathbf{a}_i^\top \boldsymbol{\phi}_t)^2 &= \frac{\left(\mathbf{a}_i^\top (\mathbf{T}\mathbf{T}^\top)^{\frac{t-1}{2}}\mathbf{T}\boldsymbol{\psi}_0\right)^2}{\left\|(\mathbf{T}\mathbf{T}^\top)^{\frac{t-1}{2}}\mathbf{T}\boldsymbol{\psi}_0\right\|^2} = \frac{\left((\mathbf{T}^\top\mathbf{a}_i)^\top \mathbf{N}^{\frac{t-1}{2}}\boldsymbol{\psi}_0\right)^2}{\boldsymbol{\psi}_0^\top \mathbf{N}^t \boldsymbol{\psi}_0} = \frac{\left(\rho_i^t \mathbf{b}_i^\top \boldsymbol{\psi}_0\right)^2}{\sum_{j=1}^r \rho_j^{2t}(\mathbf{b}_j^\top \boldsymbol{\psi}_0)^2} \\
&\leq \frac{\left(\mathbf{b}_i^\top \boldsymbol{\psi}_0\right)^2}{(\mathbf{b}_1^\top \boldsymbol{\psi}_0)^2} \exp\left(-\frac{\rho_1^2 - \rho_i^2}{\rho_1^2}t\right).
\end{aligned}
$$

Given $\delta \in (0, 1)$, define $S(\delta) = \{i : \rho_i^2 > (1 - \delta)\rho_1^2\}$. For $\delta_1, \delta_2 \in (0, 1)$, define

$$T(\delta_1, \delta_2) := \left\lceil \frac{1}{\delta_1} \log\left(\frac{1}{\mu\delta_2}\right)\right\rceil.$$

For all $i \notin S(\delta_1)$, when $t > T(\delta_1, \delta_2)$, it holds that $(\mathbf{a}_i^\top \boldsymbol{\phi}_t)^2 \leq \delta_2(\mathbf{a}_i^\top \boldsymbol{\phi}_0)^2$ if $t$ is even, and $(\mathbf{a}_i^\top \boldsymbol{\phi}_t)^2 \leq \delta_2(\mathbf{b}_i^\top \boldsymbol{\psi}_0)^2$ if $t$ is odd. In both cases, we have $\sum_{i \in S(\delta_1)}(\mathbf{a}_i^\top \boldsymbol{\phi}_t)^2 \geq 1 - \delta_2$.

When there exists a postive singular value gap, i.e., $\rho_1 - \rho_2 > 0$, set $\delta_1 = (\rho_1^2 - \rho_2^2)/\rho_1^2$ and thus $S(\delta_1) = 1$. Futhermore, set $\delta_2 = \eta$ and we obtain $(\mathbf{a}_1^\top \boldsymbol{\phi}_t)^2 \geq 1 - \eta$.

The proof for $\psi_t$ is completely analogous. To obtain the bound on the objective, we have

$$\mathbf{u}_t^\top \boldsymbol{\Sigma}_{xy} \mathbf{v}_t = \boldsymbol{\phi}_t^\top \mathbf{T} \boldsymbol{\psi}_t = \rho_1 (\boldsymbol{\phi}_t^\top \mathbf{a}_1)(\boldsymbol{\psi}_t^\top \mathbf{b}_1) + \sum_{i=2}^{r} \rho_i (\boldsymbol{\phi}_t^\top \mathbf{a}_i)(\boldsymbol{\psi}_t^\top \mathbf{b}_i)$$

$$\geq \rho_1 (\boldsymbol{\phi}_t^\top \mathbf{a}_1)(\boldsymbol{\psi}_t^\top \mathbf{b}_1) - \rho_1 \sum_{i=2}^{r} \left| \boldsymbol{\phi}_t^\top \mathbf{a}_i \right| \left| \boldsymbol{\psi}_t^\top \mathbf{b}_i \right|$$

$$\geq \rho_1 (1 - \eta) - \rho_1 \sqrt{\sum_{i=2}^{r} \left( \boldsymbol{\phi}_t^\top \mathbf{a}_i \right)^2} \sqrt{\sum_{i=2}^{r} \left( \boldsymbol{\psi}_t^\top \mathbf{b}_i \right)^2}$$

$$\geq \rho_1 (1 - \eta) - \rho_1 \eta = \rho_1 (1 - 2\eta),$$

where we have used the Cauchy-Schwarz inequality in the second inequality. $\qquad\square$

## B  Proof of Theorem 2

From now on, we distinguish the iterates of our stochastic algorithm (Algorithm 2) from the iterates of the exact power iterations (Algorithm 1) and denote the latter with asterisks, i.e., $\tilde{\mathbf{u}}_t^*$ and $\tilde{\mathbf{v}}_t^*$ for the unnormalized iterates and $\mathbf{u}_t^*$ and $\mathbf{v}_t^*$ for the normalized iterates. We denote the exact optimum of $f_t(\mathbf{u})$ and $g_t(\mathbf{v})$ by $\bar{\mathbf{u}}_t$ and $\bar{\mathbf{v}}_t$ respectively.

The following lemma bounds the distance between the iterates of inexact and exact power iterations.
***Lemma* 6.** Assume that Algorithm 1 and Algorithm 2 start with the same initialization, i.e., $\tilde{\mathbf{u}}_0 = \tilde{\mathbf{u}}_0^*$ and $\tilde{\mathbf{v}}_0 = \tilde{\mathbf{v}}_0^*$. Then, for $t \geq 1$, the unnormalized iterates of Algorithm 2 satisfy

$$\max \left( \left\| \boldsymbol{\Sigma}_{xx}^{\frac{1}{2}} \tilde{\mathbf{u}}_t - \boldsymbol{\Sigma}_{xx}^{\frac{1}{2}} \tilde{\mathbf{u}}_t^* \right\|, \left\| \boldsymbol{\Sigma}_{yy}^{\frac{1}{2}} \tilde{\mathbf{v}}_t - \boldsymbol{\Sigma}_{yy}^{\frac{1}{2}} \tilde{\mathbf{v}}_t^* \right\| \right) \leq \tilde{S}_t,$$

where

$$\tilde{S}_t := \sqrt{2\epsilon} \, \frac{(2\rho_1 / \rho_r)^t - 1}{(2\rho_1 / \rho_r) - 1}.$$

Furthermore, for $t \geq 1$, the normalized iterates of Algorithm 2 satisfy

$$\max \left( \left\| \boldsymbol{\Sigma}_{xx}^{\frac{1}{2}} \mathbf{u}_t - \boldsymbol{\Sigma}_{xx}^{\frac{1}{2}} \mathbf{u}_t^* \right\|, \left\| \boldsymbol{\Sigma}_{yy}^{\frac{1}{2}} \mathbf{v}_t - \boldsymbol{\Sigma}_{yy}^{\frac{1}{2}} \mathbf{v}_t^* \right\| \right) \leq S_t := \frac{2 \tilde{S}_t}{\rho_r}.$$

*Proof.* We focus on the $\{\tilde{\mathbf{u}}_t\}_{t \geq 0}$ and $\{\mathbf{u}_t\}_{t \geq 0}$ sequences below; the proof for $\{\tilde{\mathbf{v}}_t\}_{t \geq 0}$ and $\{\mathbf{v}_t\}_{t \geq 0}$ is completely analogous.

We prove the bound for unnormalized iterates by induction. First, the case for $t = 1$ holds trivially. For $t \geq 2$, we can bound the error of the unnormalized iterates using the exact solution to $f_t(\mathbf{u})$:

$$\left\| \boldsymbol{\Sigma}_{xx}^{\frac{1}{2}} \tilde{\mathbf{u}}_t - \boldsymbol{\Sigma}_{xx}^{\frac{1}{2}} \tilde{\mathbf{u}}_t^* \right\| \leq \left\| \boldsymbol{\Sigma}_{xx}^{\frac{1}{2}} \tilde{\mathbf{u}}_t - \boldsymbol{\Sigma}_{xx}^{\frac{1}{2}} \bar{\mathbf{u}}_t \right\| + \left\| \boldsymbol{\Sigma}_{xx}^{\frac{1}{2}} \bar{\mathbf{u}}_t - \boldsymbol{\Sigma}_{xx}^{\frac{1}{2}} \tilde{\mathbf{u}}_t^* \right\|. \tag{14}$$

For the first term of (14), notice $f_t(\mathbf{u})$ is a quadratic function with minimum achieved at $\bar{\mathbf{u}}_t = \boldsymbol{\Sigma}_{xx}^{-1} \boldsymbol{\Sigma}_{xy} \mathbf{v}_{t-1}$. For the approximate solution $\tilde{\mathbf{u}}_t$, we have

$$f_t(\tilde{\mathbf{u}}_t) - f_t(\bar{\mathbf{u}}_t) = \frac{1}{2}(\tilde{\mathbf{u}}_t - \bar{\mathbf{u}}_t)^\top \boldsymbol{\Sigma}_{xx} (\tilde{\mathbf{u}}_t - \bar{\mathbf{u}}_t) = \frac{1}{2} \left\| \boldsymbol{\Sigma}_{xx}^{\frac{1}{2}} \tilde{\mathbf{u}}_t - \boldsymbol{\Sigma}_{xx}^{\frac{1}{2}} \bar{\mathbf{u}}_t \right\|^2 \leq \epsilon.$$

It then follows that $\left\| \boldsymbol{\Sigma}_{xx}^{\frac{1}{2}} \tilde{\mathbf{u}}_t - \boldsymbol{\Sigma}_{xx}^{\frac{1}{2}} \bar{\mathbf{u}}_t \right\| \leq \sqrt{2\epsilon}$.

The second term of (14) is concerned with the error due to inexact target in the least squares problem $f_t(\mathbf{u})$ as $\mathbf{v}_{t-1}$ is different from $\mathbf{v}_{t-1}^*$. We can bound it as

$$\left\| \boldsymbol{\Sigma}_{xx}^{\frac{1}{2}} \bar{\mathbf{u}}_t - \boldsymbol{\Sigma}_{xx}^{\frac{1}{2}} \tilde{\mathbf{u}}_t^* \right\| = \left\| \boldsymbol{\Sigma}_{xx}^{\frac{1}{2}} \boldsymbol{\Sigma}_{xx}^{-1} \boldsymbol{\Sigma}_{xy} \mathbf{v}_{t-1} - \boldsymbol{\Sigma}_{xx}^{\frac{1}{2}} \boldsymbol{\Sigma}_{xx}^{-1} \boldsymbol{\Sigma}_{xy} \mathbf{v}_{t-1}^* \right\|$$

$$= \left\| \left( \boldsymbol{\Sigma}_{xx}^{-\frac{1}{2}} \boldsymbol{\Sigma}_{xy} \boldsymbol{\Sigma}_{yy}^{-\frac{1}{2}} \right) \left( \boldsymbol{\Sigma}_{yy}^{\frac{1}{2}} (\mathbf{v}_{t-1} - \mathbf{v}_{t-1}^*) \right) \right\|$$

$$\leq \| \mathbf{T} \| \left\| \boldsymbol{\Sigma}_{yy}^{\frac{1}{2}} \mathbf{v}_{t-1} - \boldsymbol{\Sigma}_{yy}^{\frac{1}{2}} \mathbf{v}_{t-1}^* \right\| = \rho_1 \left\| \boldsymbol{\Sigma}_{yy}^{\frac{1}{2}} \mathbf{v}_{t-1} - \boldsymbol{\Sigma}_{yy}^{\frac{1}{2}} \mathbf{v}_{t-1}^* \right\|. \tag{15}$$

In view of the update rule of our algorithm and the triangle inequality, we have

$$\left\|\mathbf{\Sigma}_{yy}^{\frac{1}{2}}\mathbf{v}_{t-1} - \mathbf{\Sigma}_{yy}^{\frac{1}{2}}\mathbf{v}_{t-1}^*\right\|$$

$$\leq \left\|\frac{\mathbf{\Sigma}_{yy}^{\frac{1}{2}}\tilde{\mathbf{v}}_{t-1}}{\left\|\mathbf{\Sigma}_{yy}^{\frac{1}{2}}\tilde{\mathbf{v}}_{t-1}\right\|} - \frac{\mathbf{\Sigma}_{yy}^{\frac{1}{2}}\tilde{\mathbf{v}}_{t-1}}{\left\|\mathbf{\Sigma}_{yy}^{\frac{1}{2}}\tilde{\mathbf{v}}_{t-1}^*\right\|}\right\| + \left\|\frac{\mathbf{\Sigma}_{yy}^{\frac{1}{2}}\tilde{\mathbf{v}}_{t-1}}{\left\|\mathbf{\Sigma}_{yy}^{\frac{1}{2}}\tilde{\mathbf{v}}_{t-1}^*\right\|} - \frac{\mathbf{\Sigma}_{yy}^{\frac{1}{2}}\tilde{\mathbf{v}}_{t-1}^*}{\left\|\mathbf{\Sigma}_{yy}^{\frac{1}{2}}\tilde{\mathbf{v}}_{t-1}^*\right\|}\right\|$$

$$= \left\|\mathbf{\Sigma}_{yy}^{\frac{1}{2}}\tilde{\mathbf{v}}_{t-1}\right\| \left|\frac{1}{\left\|\mathbf{\Sigma}_{yy}^{\frac{1}{2}}\tilde{\mathbf{v}}_{t-1}\right\|} - \frac{1}{\left\|\mathbf{\Sigma}_{yy}^{\frac{1}{2}}\tilde{\mathbf{v}}_{t-1}^*\right\|}\right| + \frac{1}{\left\|\mathbf{\Sigma}_{yy}^{\frac{1}{2}}\tilde{\mathbf{v}}_{t-1}^*\right\|}\left\|\mathbf{\Sigma}_{yy}^{\frac{1}{2}}\tilde{\mathbf{v}}_{t-1} - \mathbf{\Sigma}_{yy}^{\frac{1}{2}}\tilde{\mathbf{v}}_{t-1}^*\right\|$$

$$= \frac{1}{\left\|\mathbf{\Sigma}_{yy}^{\frac{1}{2}}\tilde{\mathbf{v}}_{t-1}^*\right\|}\left|\left\|\mathbf{\Sigma}_{yy}^{\frac{1}{2}}\tilde{\mathbf{v}}_{t-1}^*\right\| - \left\|\mathbf{\Sigma}_{yy}^{\frac{1}{2}}\tilde{\mathbf{v}}_{t-1}\right\|\right| + \frac{1}{\left\|\mathbf{\Sigma}_{yy}^{\frac{1}{2}}\tilde{\mathbf{v}}_{t-1}^*\right\|}\left\|\mathbf{\Sigma}_{yy}^{\frac{1}{2}}\tilde{\mathbf{v}}_{t-1} - \mathbf{\Sigma}_{yy}^{\frac{1}{2}}\tilde{\mathbf{v}}_{t-1}^*\right\|$$

$$\leq \frac{2}{\left\|\mathbf{\Sigma}_{yy}^{\frac{1}{2}}\tilde{\mathbf{v}}_{t-1}^*\right\|}\left\|\mathbf{\Sigma}_{yy}^{\frac{1}{2}}\tilde{\mathbf{v}}_{t-1} - \mathbf{\Sigma}_{yy}^{\frac{1}{2}}\tilde{\mathbf{v}}_{t-1}^*\right\| \leq \frac{2\tilde{S}_{t-1}}{\left\|\mathbf{\Sigma}_{yy}^{\frac{1}{2}}\tilde{\mathbf{v}}_{t-1}^*\right\|}. \tag{16}$$

We now bound $\left\|\mathbf{\Sigma}_{yy}^{\frac{1}{2}}\tilde{\mathbf{v}}_{t-1}^*\right\|$ from below. Since $t \geq 2$, we have

$$\mathbf{\Sigma}_{yy}^{\frac{1}{2}}\tilde{\mathbf{v}}_{t-1}^* = \mathbf{\Sigma}_{yy}^{\frac{1}{2}}\mathbf{\Sigma}_{yy}^{-1}\mathbf{\Sigma}_{xy}^{\top}\mathbf{u}_{t-2}^* = \left(\mathbf{\Sigma}_{yy}^{-\frac{1}{2}}\mathbf{\Sigma}_{xy}^{\top}\mathbf{\Sigma}_{xx}^{-\frac{1}{2}}\right)\left(\mathbf{\Sigma}_{xx}^{\frac{1}{2}}\mathbf{u}_{t-2}^*\right) = \mathbf{T}^{\top}\left(\mathbf{\Sigma}_{xx}^{\frac{1}{2}}\mathbf{u}_{t-2}^*\right).$$

Now, $\mathbf{\Sigma}_{xx}^{\frac{1}{2}}\mathbf{u}_{t-2}^*$ corresponds to $\phi_{t-2}$ in Algorithm 1, which has unit length and lies in the span of $\{\mathbf{a}_1,\ldots,\mathbf{a}_r\}$, so we have

$$\left\|\mathbf{\Sigma}_{yy}^{\frac{1}{2}}\tilde{\mathbf{v}}_{t-1}^*\right\| = \left\|\mathbf{T}^{\top}\phi_{t-2}\right\| \geq \rho_r.$$

Combining (14), (15) and (16) gives

$$\left\|\mathbf{\Sigma}_{xx}^{\frac{1}{2}}\tilde{\mathbf{u}}_t - \mathbf{\Sigma}_{xx}^{\frac{1}{2}}\tilde{\mathbf{u}}_t^*\right\| \leq \sqrt{2\epsilon} + \frac{2\rho_1}{\rho_r} \cdot \tilde{S}_{t-1} = \sqrt{2\epsilon} + \frac{2\rho_1}{\rho_r} \cdot \sqrt{2\epsilon}\frac{(2\rho_1/\rho_r)^{t-1} - 1}{(2\rho_1/\rho_r) - 1}$$

$$= \sqrt{2\epsilon}\frac{(2\rho_1/\rho_r)^t - 1}{(2\rho_1/\rho_r) - 1} = \tilde{S}_t.$$

The bound for normalized iterates follows from (16). $\qquad\square$

***Proof of Theorem 2.*** We prove the theorem by relating the iterates of inexact power iterations to those of exact power iterations.

Assume the same initialization as in Lemma 6. First observe that

$$(\mathbf{u}_t^{\top}\mathbf{\Sigma}_{xx}\mathbf{u}^*)^2 = \left((\mathbf{u}_t^*)^{\top}\mathbf{\Sigma}_{xx}\mathbf{u}^* + (\mathbf{u}_t - \mathbf{u}_t^*)^{\top}\mathbf{\Sigma}_{xx}\mathbf{u}^*\right)^2$$

$$\geq \left((\mathbf{u}_t^*)^{\top}\mathbf{\Sigma}_{xx}\mathbf{u}^*\right)^2 + 2\left((\mathbf{u}_t^*)^{\top}\mathbf{\Sigma}_{xx}\mathbf{u}^*\right)\left((\mathbf{u}_t - \mathbf{u}_t^*)^{\top}\mathbf{\Sigma}_{xx}\mathbf{u}^*\right)$$

$$\geq \left((\mathbf{u}_t^*)^{\top}\mathbf{\Sigma}_{xx}\mathbf{u}^*\right)^2 - 2\left|\left(\mathbf{\Sigma}_{xx}^{\frac{1}{2}}(\mathbf{u}_t - \mathbf{u}_t^*)\right)^{\top}\left(\mathbf{\Sigma}_{xx}^{\frac{1}{2}}\mathbf{u}^*\right)\right|$$

$$\geq \left((\mathbf{u}_t^*)^{\top}\mathbf{\Sigma}_{xx}\mathbf{u}^*\right)^2 - 2\left\|\mathbf{\Sigma}_{xx}^{\frac{1}{2}}\mathbf{u}_t - \mathbf{\Sigma}_{xx}^{\frac{1}{2}}\mathbf{u}_t^*\right\| \tag{17}$$

where we have used the fact that $\left\|\mathbf{\Sigma}_{xx}^{\frac{1}{2}}\mathbf{u}_t\right\| = \left\|\mathbf{\Sigma}_{xx}^{\frac{1}{2}}\mathbf{u}_t^*\right\| = \left\|\mathbf{\Sigma}_{xx}^{\frac{1}{2}}\mathbf{u}^*\right\| = 1$ and the Cauchy-Schwarz inequality in the last two steps.

Applying Theorem 1 with $T \geq \lceil\frac{\rho_1^2}{\rho_1^2-\rho_2^2}\log\left(\frac{2}{\mu\eta}\right)\rceil$, we have that $\left((\mathbf{u}_T^*)^{\top}\mathbf{\Sigma}_{xx}\mathbf{u}^*\right)^2 \geq 1 - \eta/2$. On the other hand, in view of Lemma 6, we have for the specified $\epsilon$ value in Algorithm 2 that $\left\|\mathbf{\Sigma}_{xx}^{\frac{1}{2}}\mathbf{u}_T - \mathbf{\Sigma}_{xx}^{\frac{1}{2}}\mathbf{u}_T^*\right\| \leq S_T = \eta/4$. Plugging these two bounds into (17) gives the desired result.

The proof for $\mathbf{v}_T$ is completely analogous. $\qquad\square$

## C   SVRG for minimizing $f(\mathbf{u})$

We provide the pseudo-code of SVRG for solving the least squares problem (6) below.

---

SVRG for $\min_{\mathbf{u}}\ f(\mathbf{u}) := \frac{1}{N}\sum_{i=1}^{N}\left(\frac{1}{2}\left|\mathbf{u}^{\top}\mathbf{x}_i - \mathbf{v}^{\top}\mathbf{y}_i\right|^2 + \frac{\gamma_x}{2}\|\mathbf{u}\|^2\right).$

---

**Input:** Stepsize $\xi$.
  Initialize $\mathbf{u}_{(0)} \in \mathbb{R}^{d_x}$.
  **for** $j = 1, 2, \ldots, M$ **do**
    $\mathbf{w}_0 \leftarrow \mathbf{u}_{(j-1)}$
    Evaluate the batch gradient $\nabla f(\mathbf{w}_0) = \mathbf{X}(\mathbf{X}^{\top}\mathbf{w}_0 - \mathbf{Y}^{\top}\mathbf{v})/N + \gamma_x \mathbf{w}_0$
    **for** $t = 1, 2, \ldots, m$ **do**
      Randomly pick $i_t$ from $\{1, \ldots, N\}$
      $\mathbf{w}_t \leftarrow \mathbf{w}_{t-1} - \xi\left((\mathbf{x}_{i_t}\mathbf{x}_{i_t}^{\top} + \gamma_x\mathbf{I})(\mathbf{w}_{t-1} - \mathbf{w}_0) + \nabla f(\mathbf{w}_0)\right)$
    **end for**
    $\mathbf{u}_{(j)} \leftarrow \mathbf{w}_t$ for randomly chosen $t \in \{1, \ldots, m\}$.
  **end for**
**Output:** $\mathbf{u}_{(M)}$ is the approximate solution.

---

## D   Initial suboptimality of warm-starts in Algorithm 2

At time step $t$, we initialize the least squares problem $f_t(\mathbf{u})$ with the unnormalized iterate $\tilde{\mathbf{u}}_{t-1}$ from the previous time step. We now bound the suboptimality of this initialization. Observe that the minimum of $f_t(\mathbf{u})$ is achieved by $\bar{\mathbf{u}}_t = \mathbf{\Sigma}_{xx}^{-1}\mathbf{\Sigma}_{xy}\mathbf{v}_{t-1}$, and that

$$f_t(\tilde{\mathbf{u}}_{t-1}) - f_t(\bar{\mathbf{u}}_t) = \frac{1}{2}(\tilde{\mathbf{u}}_{t-1} - \bar{\mathbf{u}}_t)^{\top}\mathbf{\Sigma}_{xx}(\tilde{\mathbf{u}}_{t-1} - \bar{\mathbf{u}}_t) = \frac{1}{2}\left\|\mathbf{\Sigma}_{xx}^{\frac{1}{2}}\tilde{\mathbf{u}}_{t-1} - \mathbf{\Sigma}_{xx}^{\frac{1}{2}}\bar{\mathbf{u}}_t\right\|^2.$$

Applying the triangle inequality, we have for $t = 1$ that

$$\left\|\mathbf{\Sigma}_{xx}^{\frac{1}{2}}\tilde{\mathbf{u}}_0 - \mathbf{\Sigma}_{xx}^{\frac{1}{2}}\bar{\mathbf{u}}_1\right\| \le \left\|\mathbf{\Sigma}_{xx}^{\frac{1}{2}}\tilde{\mathbf{u}}_0\right\| + \left\|\mathbf{\Sigma}_{xx}^{\frac{1}{2}}\bar{\mathbf{u}}_1\right\| \le 1 + \left\|\mathbf{\Sigma}_{xx}^{\frac{1}{2}}\mathbf{\Sigma}_{xx}^{-1}\mathbf{\Sigma}_{xy}\mathbf{v}_0\right\|$$

$$= 1 + \left\|\mathbf{T}\mathbf{\Sigma}_{yy}^{\frac{1}{2}}\mathbf{v}_0\right\| \le 1 + \|\mathbf{T}\|\left\|\mathbf{\Sigma}_{yy}^{\frac{1}{2}}\mathbf{v}_0\right\| = 1 + \rho_1 \le 2$$

where we have used facts that $\left\|\mathbf{\Sigma}_{yy}^{\frac{1}{2}}\tilde{\mathbf{u}}_0\right\| = \left\|\mathbf{\Sigma}_{yy}^{\frac{1}{2}}\mathbf{v}_0\right\| = 1$ due to the initial normalizations.

And we have for $t \ge 2$ that

$$\left\|\mathbf{\Sigma}_{xx}^{\frac{1}{2}}\tilde{\mathbf{u}}_{t-1} - \mathbf{\Sigma}_{xx}^{\frac{1}{2}}\bar{\mathbf{u}}_t\right\| \le \left\|\mathbf{\Sigma}_{xx}^{\frac{1}{2}}\tilde{\mathbf{u}}_{t-1} - \mathbf{\Sigma}_{xx}^{\frac{1}{2}}\bar{\mathbf{u}}_{t-1}\right\| + \left\|\mathbf{\Sigma}_{xx}^{\frac{1}{2}}\bar{\mathbf{u}}_{t-1} - \mathbf{\Sigma}_{xx}^{\frac{1}{2}}\bar{\mathbf{u}}_t\right\|$$

$$\le \sqrt{2\epsilon} + \left\|\mathbf{\Sigma}_{xx}^{\frac{1}{2}}\mathbf{\Sigma}_{xx}^{-1}\mathbf{\Sigma}_{xy}\mathbf{v}_{t-2} - \mathbf{\Sigma}_{xx}^{\frac{1}{2}}\mathbf{\Sigma}_{xx}^{-1}\mathbf{\Sigma}_{xy}\mathbf{v}_{t-1}\right\|$$

$$= \sqrt{2\epsilon} + \left\|\mathbf{T}\left(\mathbf{\Sigma}_{yy}^{\frac{1}{2}}\mathbf{v}_{t-2} - \mathbf{\Sigma}_{yy}^{\frac{1}{2}}\mathbf{v}_{t-1}\right)\right\|$$

$$\le \sqrt{2\epsilon} + \|\mathbf{T}\|\left\|\mathbf{\Sigma}_{yy}^{\frac{1}{2}}\mathbf{v}_{t-2} - \mathbf{\Sigma}_{yy}^{\frac{1}{2}}\mathbf{v}_{t-1}\right\|$$

$$\le \sqrt{2\epsilon} + 2\rho_1 \le \sqrt{2\epsilon} + 2$$

where we have used the fact that $\left\|\mathbf{\Sigma}_{yy}^{\frac{1}{2}}\mathbf{v}_{t-2}\right\| = \left\|\mathbf{\Sigma}_{yy}^{\frac{1}{2}}\mathbf{v}_{t-1}\right\| = 1$ in the last inequality.

Therefore, for all $t \ge 1$, the ration between initial suboptimality and required accuracy is

$$\frac{f_t(\tilde{\mathbf{u}}_{t-1}) - f_t(\bar{\mathbf{u}}_t)}{\epsilon} \sim \frac{2}{\epsilon}.$$

# E  The shift-and-invert preconditioning (SI) algorithm for CCA

Our shift-and-invert preconditioning (SI) meta-algorithm is detailed in Algorithm 3.

---

**Algorithm 3** The shift-and-invert preconditioning meta-algorithm for CCA.

---

**Input:** Data matrices $\mathbf{X}$, $\mathbf{Y}$, regularization parameters $(\gamma_x, \gamma_y)$, an estimate $\tilde{\Delta}$ for $\Delta = \rho_1 - \rho_2$.

Initialize $\tilde{\mathbf{u}}_0 \in \mathbb{R}^{d_x}$, $\tilde{\mathbf{v}}_0 \in \mathbb{R}^{d_y}$

$\mathbf{u}_0 \leftarrow \tilde{\mathbf{u}}_0 / \sqrt{\tilde{\mathbf{u}}_0^\top \mathbf{\Sigma}_{xx} \tilde{\mathbf{u}}_0}, \qquad \mathbf{v}_0 \leftarrow \tilde{\mathbf{v}}_0 / \sqrt{\tilde{\mathbf{v}}_0^\top \mathbf{\Sigma}_{yy} \tilde{\mathbf{v}}_0}$

**// Phase I: shift-and-invert preconditioning for eigenvectors of $\mathbf{M}_\lambda$**

$s \leftarrow 0, \qquad \lambda_{(0)} \leftarrow 1 + \tilde{\Delta}$

**repeat**

  $s \leftarrow s + 1$

  **for** $t = (s-1)m_1 + 1, \ldots, sm_1$ **do**

    Optimize the least squares problem

$$\min_{\mathbf{u},\mathbf{v}} \; h_t(\mathbf{u}, \mathbf{v}) := \frac{1}{2} \begin{bmatrix} \mathbf{u}^\top \mathbf{v}^\top \end{bmatrix} \begin{bmatrix} \lambda_{(s-1)} \mathbf{\Sigma}_{xx} & -\mathbf{\Sigma}_{xy} \\ -\mathbf{\Sigma}_{xy}^\top & \lambda_{(s-1)} \mathbf{\Sigma}_{yy} \end{bmatrix} \begin{bmatrix} \mathbf{u} \\ \mathbf{v} \end{bmatrix} - \mathbf{u}^\top \mathbf{\Sigma}_{xx} \mathbf{u}_{t-1} - \mathbf{v}^\top \mathbf{\Sigma}_{yy} \mathbf{v}_{t-1}$$

    and output an approximate solution $(\tilde{\mathbf{u}}_t, \tilde{\mathbf{v}}_t)$ satisfying $h_t(\tilde{\mathbf{u}}_t, \tilde{\mathbf{v}}_t) \leq \min_{\mathbf{u},\mathbf{v}} h_t(\mathbf{u}, \mathbf{v}) + \tilde{\epsilon}$.

    Normalization: $\begin{bmatrix} \mathbf{u}_t \\ \mathbf{v}_t \end{bmatrix} \leftarrow \sqrt{2} \begin{bmatrix} \tilde{\mathbf{u}}_t \\ \tilde{\mathbf{v}}_t \end{bmatrix} \Big/ \sqrt{\tilde{\mathbf{u}}_t^\top \mathbf{\Sigma}_{xx} \tilde{\mathbf{u}}_t + \tilde{\mathbf{v}}_t^\top \mathbf{\Sigma}_{yy} \tilde{\mathbf{v}}_t}$

  **end for**

  Optimize the least squares problem

$$\min_{\mathbf{w}} \; l_s(\mathbf{w}) := \frac{1}{2} \mathbf{w}^\top \begin{bmatrix} \lambda_{(s-1)} \mathbf{\Sigma}_{xx} & -\mathbf{\Sigma}_{xy} \\ -\mathbf{\Sigma}_{xy}^\top & \lambda_{(s-1)} \mathbf{\Sigma}_{yy} \end{bmatrix} \mathbf{w} - \mathbf{w}^\top \begin{bmatrix} \mathbf{\Sigma}_{xx} \mathbf{u}_{sm_1} \\ \mathbf{\Sigma}_{yy} \mathbf{v}_{sm_1} \end{bmatrix}$$

  and output an approximate solution $\mathbf{w}_s$ satisfying $l_s(\mathbf{w}_s) \leq \min_{\mathbf{w}} l_s(\mathbf{w}) + \tilde{\epsilon}$.

  $\Delta_s \leftarrow \frac{1}{2} \cdot \dfrac{1}{\frac{1}{2}\begin{bmatrix} \mathbf{u}_{sm_1}^\top \mathbf{v}_{sm_1}^\top \end{bmatrix} \begin{bmatrix} \mathbf{\Sigma}_{xx} & \\ & \mathbf{\Sigma}_{yy} \end{bmatrix} \mathbf{w}_s - 2\sqrt{\tilde{\epsilon}/\tilde{\Delta}}}, \qquad \lambda_{(s)} \leftarrow \lambda_{(s-1)} - \frac{\Delta_s}{2}$

**until** $\Delta_{(s)} \leq \tilde{\Delta}$

$\lambda_{(f)} \leftarrow \lambda_{(s)}$

**for** $t = sm_1 + 1, sm_1 + 2, \ldots, sm_1 + m_2$ **do**

  Optimize the least squares problem

$$\min_{\mathbf{u},\mathbf{v}} \; h_t(\mathbf{u}, \mathbf{v}) := \frac{1}{2} \begin{bmatrix} \mathbf{u}^\top \mathbf{v}^\top \end{bmatrix} \begin{bmatrix} \lambda_{(f)} \mathbf{\Sigma}_{xx} & -\mathbf{\Sigma}_{xy} \\ -\mathbf{\Sigma}_{xy}^\top & \lambda_{(f)} \mathbf{\Sigma}_{yy} \end{bmatrix} \begin{bmatrix} \mathbf{u} \\ \mathbf{v} \end{bmatrix} - \mathbf{u}^\top \mathbf{\Sigma}_{xx} \mathbf{u}_{t-1} - \mathbf{v}^\top \mathbf{\Sigma}_{yy} \mathbf{v}_{t-1}$$

  and output an approximate solution $(\tilde{\mathbf{u}}_t, \tilde{\mathbf{v}}_t)$ satisfying $h_t(\tilde{\mathbf{u}}_t, \tilde{\mathbf{v}}_t) \leq \min_{\mathbf{u},\mathbf{v}} h_t(\mathbf{u}, \mathbf{v}) + \tilde{\epsilon}$.

  Normalization: $\begin{bmatrix} \mathbf{u}_t \\ \mathbf{v}_t \end{bmatrix} \leftarrow \sqrt{2} \begin{bmatrix} \tilde{\mathbf{u}}_t \\ \tilde{\mathbf{v}}_t \end{bmatrix} \Big/ \sqrt{\tilde{\mathbf{u}}_t^\top \mathbf{\Sigma}_{xx} \tilde{\mathbf{u}}_t + \tilde{\mathbf{v}}_t^\top \mathbf{\Sigma}_{yy} \tilde{\mathbf{v}}_t}$

**end for**

**// Phase II: Final normalization**

$T \leftarrow sm_1 + m_2, \qquad \hat{\mathbf{u}} \leftarrow \mathbf{u}_T / \sqrt{\mathbf{u}_T^\top \mathbf{\Sigma}_{xx} \mathbf{u}_T}, \qquad \hat{\mathbf{v}} \leftarrow \mathbf{v}_T / \sqrt{\mathbf{v}_T^\top \mathbf{\Sigma}_{yy} \mathbf{v}_T}$

**Output:** $(\hat{\mathbf{u}}, \hat{\mathbf{v}})$ is the approximate solution to CCA.

---

# F  Proof of Theorem 4

The proof of Theorem 4 closely follows that of [16, Theorem 4.2]. And we will need a few lemmas on the convergence of inexact power iterations.

## F.1  Auxiliary lemmas

Define the condition number of $\mathbf{M}_\lambda$ as

$$\kappa_\lambda := \frac{\sigma_1(\mathbf{M}_\lambda)}{\sigma_d(\mathbf{M}_\lambda)} = \frac{\frac{1}{\lambda-\rho_1}}{\frac{1}{\lambda+\rho_1}} = \frac{\lambda+\rho_1}{\lambda-\rho_1},$$

and the inverse relative spectral gap of $\mathbf{M}_\lambda$ as

$$\delta_\lambda := \frac{\sigma_1(\mathbf{M}_\lambda)}{\sigma_1(\mathbf{M}_\lambda)-\sigma_2(\mathbf{M}_\lambda)} = \frac{\frac{1}{\lambda-\rho_1}}{\frac{1}{\lambda-\rho_1}-\frac{1}{\lambda-\rho_2}} = \frac{\lambda-\rho_2}{\rho_1-\rho_2}.$$

The first lemma states the convergence of exact power iterations, paralleling [16, Theorem A.1].

***Lemma*** 7 (Convergence of exact power iterations). Fix $\alpha > 0$. For the exact power iterations on $\mathbf{M}_\lambda$ where

$$\begin{bmatrix} \tilde{\mathbf{u}}_t^* \\ \tilde{\mathbf{v}}_t^* \end{bmatrix} \leftarrow \begin{bmatrix} \lambda\boldsymbol{\Sigma}_{xx} & -\boldsymbol{\Sigma}_{xy} \\ -\boldsymbol{\Sigma}_{xy}^\top & \lambda\boldsymbol{\Sigma}_{yy} \end{bmatrix}^{-1} \begin{bmatrix} \boldsymbol{\Sigma}_{xx} & \\ & \boldsymbol{\Sigma}_{yy} \end{bmatrix} \begin{bmatrix} \mathbf{u}_{t-1}^* \\ \mathbf{v}_{t-1}^* \end{bmatrix},$$

$$\begin{bmatrix} \mathbf{u}_t^* \\ \mathbf{v}_t^* \end{bmatrix} \leftarrow \sqrt{2} \begin{bmatrix} \tilde{\mathbf{u}}_t^* \\ \tilde{\mathbf{v}}_t^* \end{bmatrix} \Big/ \sqrt{(\tilde{\mathbf{u}}_t^*)^\top \boldsymbol{\Sigma}_{xx}\tilde{\mathbf{u}}_t^* + (\tilde{\mathbf{v}}_t^*)^\top \boldsymbol{\Sigma}_{yy}\tilde{\mathbf{v}}_t^*}, \qquad \text{for} \quad t = 1,\dots,m,$$

and $\mu' := \frac{1}{4}\left((\mathbf{u}_0^*)^\top \boldsymbol{\Sigma}_{xx}\mathbf{u}^* + (\mathbf{v}_0^*)^\top \boldsymbol{\Sigma}_{yy}\mathbf{v}^*\right)^2 > 0$, we have

- (crude regime)

$$\frac{1}{2}\left[(\mathbf{u}_t^*)^\top \boldsymbol{\Sigma}_{xx}^{\frac{1}{2}}, (\mathbf{v}_t^*)^\top \boldsymbol{\Sigma}_{yy}^{\frac{1}{2}}\right] \mathbf{M}_\lambda \begin{bmatrix} \boldsymbol{\Sigma}_{xx}^{\frac{1}{2}}\mathbf{u}_t^* \\ \boldsymbol{\Sigma}_{yy}^{\frac{1}{2}}\mathbf{v}_t^* \end{bmatrix} \geq (1-\alpha)\cdot\sigma_1(\mathbf{M}_\lambda)$$

  for $t \geq \lceil \frac{1}{\alpha}\log\left(\frac{2}{\mu'\alpha}\right)\rceil$,

- (accurate regime)

$$\frac{1}{4}\left((\mathbf{u}_t^*)^\top \boldsymbol{\Sigma}_{xx}\mathbf{u}^* + (\mathbf{v}_t^*)^\top \boldsymbol{\Sigma}_{yy}\mathbf{v}^*\right)^2 \geq 1-\alpha$$

  for $t \geq \lceil \frac{\delta_\lambda}{2}\log\left(\frac{1}{\mu'\alpha}\right)\rceil$.

The second lemma bounds the distances between the iterates of inexact and exact power iterations, paralleling [16, Lemma 4.1]. Recall that the $(\tilde{\mathbf{u}}_t, \tilde{\mathbf{v}}_t)$ in Algorithm 3 satisfies $h_t(\tilde{\mathbf{u}}_t, \tilde{\mathbf{v}}_t) \leq \min_{\mathbf{u},\mathbf{v}} h_t(\mathbf{u}, \mathbf{v}) + \tilde{\epsilon}$. Let $(\bar{\mathbf{u}}_t, \bar{\mathbf{v}}_t)$ be the exact minimum of $h_t$. Then we have

$$h_t(\tilde{\mathbf{u}}_t, \tilde{\mathbf{v}}_t) - h_t(\bar{\mathbf{u}}_t, \bar{\mathbf{v}}_t)$$

$$= \frac{1}{2}\left[(\tilde{\mathbf{u}}_t - \bar{\mathbf{u}}_t)^\top \ (\tilde{\mathbf{v}}_t - \bar{\mathbf{v}}_t)^\top\right] \begin{bmatrix} \lambda\boldsymbol{\Sigma}_{xx} & -\boldsymbol{\Sigma}_{xy} \\ -\boldsymbol{\Sigma}_{xy}^\top & \lambda\boldsymbol{\Sigma}_{yy} \end{bmatrix} \begin{bmatrix} \tilde{\mathbf{u}}_t - \bar{\mathbf{u}}_t \\ \tilde{\mathbf{v}}_t - \bar{\mathbf{v}}_t \end{bmatrix}$$

$$= \frac{1}{2}\left[(\tilde{\mathbf{u}}_t - \bar{\mathbf{u}}_t)^\top \ (\tilde{\mathbf{v}}_t - \bar{\mathbf{v}}_t)^\top\right] \begin{bmatrix} \lambda\boldsymbol{\Sigma}_{xx} & -\boldsymbol{\Sigma}_{xy} \\ -\boldsymbol{\Sigma}_{xy}^\top & \lambda\boldsymbol{\Sigma}_{yy} \end{bmatrix} \begin{bmatrix} \tilde{\mathbf{u}}_t - \bar{\mathbf{u}}_t \\ \tilde{\mathbf{v}}_t - \bar{\mathbf{v}}_t \end{bmatrix}$$

$$= \frac{1}{2}\left[(\tilde{\mathbf{u}}_t - \bar{\mathbf{u}}_t)^\top\boldsymbol{\Sigma}_{xx}^{\frac{1}{2}} \ (\tilde{\mathbf{v}}_t - \bar{\mathbf{v}}_t)^\top\boldsymbol{\Sigma}_{yy}^{\frac{1}{2}}\right] \begin{bmatrix} \lambda\mathbf{I} & -\mathbf{T} \\ -\mathbf{T}^\top & \lambda\mathbf{I} \end{bmatrix} \begin{bmatrix} \boldsymbol{\Sigma}_{xx}^{\frac{1}{2}}(\tilde{\mathbf{u}}_t - \bar{\mathbf{u}}_t) \\ \boldsymbol{\Sigma}_{yy}^{\frac{1}{2}}(\tilde{\mathbf{v}}_t - \bar{\mathbf{v}}_t) \end{bmatrix}$$

$$= \frac{1}{2}\left[(\tilde{\mathbf{u}}_t - \bar{\mathbf{u}}_t)^\top\boldsymbol{\Sigma}_{xx}^{\frac{1}{2}} \ (\tilde{\mathbf{v}}_t - \bar{\mathbf{v}}_t)^\top\boldsymbol{\Sigma}_{yy}^{\frac{1}{2}}\right] \mathbf{M}_\lambda^{-1} \begin{bmatrix} \boldsymbol{\Sigma}_{xx}^{\frac{1}{2}}(\tilde{\mathbf{u}}_t - \bar{\mathbf{u}}_t) \\ \boldsymbol{\Sigma}_{yy}^{\frac{1}{2}}(\tilde{\mathbf{v}}_t - \bar{\mathbf{v}}_t) \end{bmatrix} \leq \tilde{\epsilon}. \qquad (18)$$

***Lemma*** 8 (Power iterations with inexact matrix-vector multiplications). Consider the inexact power iterations on $\mathbf{M}_\lambda$ where

$$(\tilde{\mathbf{u}}_t, \tilde{\mathbf{v}}_t) \qquad \text{satisfies} \qquad (18),$$

$$\begin{bmatrix} \mathbf{u}_t \\ \mathbf{v}_t \end{bmatrix} \leftarrow \sqrt{2} \begin{bmatrix} \tilde{\mathbf{u}}_t \\ \tilde{\mathbf{v}}_t \end{bmatrix} \Big/ \sqrt{\tilde{\mathbf{u}}_t^\top \boldsymbol{\Sigma}_{xx}\tilde{\mathbf{u}}_t + \tilde{\mathbf{v}}_t^\top \boldsymbol{\Sigma}_{yy}\tilde{\mathbf{v}}_t}, \qquad \text{for} \quad t = 1,\dots,m.$$

Compare these iterates with those of the exact power iterations described in Lemma 7 using the same initialization $\tilde{\mathbf{u}}_0 = \tilde{\mathbf{u}}_0^*$, $\tilde{\mathbf{v}}_0 = \tilde{\mathbf{v}}_0^*$. Then, for $t \geq 0$, the unnormalized iterates satisfy

$$\left\| \frac{1}{\sqrt{2}} \begin{bmatrix} \mathbf{\Sigma}_{xx}^{\frac{1}{2}} \tilde{\mathbf{u}}_t \\ \mathbf{\Sigma}_{yy}^{\frac{1}{2}} \tilde{\mathbf{v}}_t \end{bmatrix} - \frac{1}{\sqrt{2}} \begin{bmatrix} \mathbf{\Sigma}_{xx}^{\frac{1}{2}} \tilde{\mathbf{u}}_t^* \\ \mathbf{\Sigma}_{yy}^{\frac{1}{2}} \tilde{\mathbf{v}}_t^* \end{bmatrix} \right\| \leq \tilde{R}_t$$

where

$$\tilde{R}_t := \sqrt{\sigma_1(\mathbf{M}_\lambda) \cdot \tilde{\epsilon}} \cdot \frac{(2\kappa_\lambda)^t - 1}{2\kappa_\lambda - 1},$$

while the normalized iterates satisfy

$$\left\| \frac{1}{\sqrt{2}} \begin{bmatrix} \mathbf{\Sigma}_{xx}^{\frac{1}{2}} \mathbf{u}_t \\ \mathbf{\Sigma}_{yy}^{\frac{1}{2}} \mathbf{v}_t \end{bmatrix} - \frac{1}{\sqrt{2}} \begin{bmatrix} \mathbf{\Sigma}_{xx}^{\frac{1}{2}} \mathbf{u}_t^* \\ \mathbf{\Sigma}_{yy}^{\frac{1}{2}} \mathbf{v}_t^* \end{bmatrix} \right\| \leq R_t := \frac{2\tilde{R}_t}{\sigma_d(\mathbf{M}_\lambda)}.$$

The third lemma states the convergence of inexact power iterations, paralleling [16, Theorem 4.1].

**Lemma 9** (Convergence of inexact power iterations). Fix $\alpha > 0$. Consider the inexact power iterations described in Lemma 8.

- (crude regime) Let $t_1 = \lceil \frac{2}{\alpha} \log\left(\frac{4}{\mu'\alpha}\right) \rceil$. Fix $T \geq t_1$, and set $\tilde{\epsilon}(T) = \frac{\alpha^2 \cdot \sigma_d(\mathbf{M}_\lambda)}{64\kappa_\lambda} \left(\frac{2\kappa_\lambda - 1}{(2\kappa_\lambda)^T - 1}\right)^2$. Then we have

$$\frac{1}{2} \left[ \mathbf{u}_T^\top \mathbf{\Sigma}_{xx}^{\frac{1}{2}}, \ \mathbf{v}_T^\top \mathbf{\Sigma}_{yy}^{\frac{1}{2}} \right] \mathbf{M}_\lambda \begin{bmatrix} \mathbf{\Sigma}_{xx}^{\frac{1}{2}} \mathbf{u}_T \\ \mathbf{\Sigma}_{yy}^{\frac{1}{2}} \mathbf{v}_T \end{bmatrix} \geq (1 - \alpha) \cdot \sigma_1(\mathbf{M}_\lambda).$$

- (accurate regime) Let $t_2 = \lceil \frac{\delta(\mathbf{M}_\lambda)}{2} \log\left(\frac{2}{\mu'\alpha}\right) \rceil$. Fix $T \geq t_2$, and set $\tilde{\epsilon}(T) = \frac{\alpha^2 \cdot \sigma_d(\mathbf{M}_\lambda)}{64\kappa_\lambda} \left(\frac{2\kappa_\lambda - 1}{(2\kappa_\lambda)^T - 1}\right)^2$. Then we have

$$\frac{1}{4} \left( \mathbf{u}_T^\top \mathbf{\Sigma}_{xx} \mathbf{u}^* + \mathbf{v}_T^\top \mathbf{\Sigma}_{yy} \mathbf{v}^* \right)^2 \geq 1 - \alpha.$$

For brevity, let us define the following short-hands:

$$\tilde{\mathbf{r}}_t = \frac{1}{\sqrt{2}} \begin{bmatrix} \mathbf{\Sigma}_{xx}^{\frac{1}{2}} \tilde{\mathbf{u}}_t \\ \mathbf{\Sigma}_{yy}^{\frac{1}{2}} \tilde{\mathbf{v}}_t \end{bmatrix}, \qquad \mathbf{r}_t = \frac{1}{\sqrt{2}} \begin{bmatrix} \mathbf{\Sigma}_{xx}^{\frac{1}{2}} \mathbf{u}_t \\ \mathbf{\Sigma}_{yy}^{\frac{1}{2}} \mathbf{v}_t \end{bmatrix}, \qquad \bar{\mathbf{r}}_t = \frac{1}{\sqrt{2}} \begin{bmatrix} \mathbf{\Sigma}_{xx}^{\frac{1}{2}} \bar{\mathbf{u}}_t \\ \mathbf{\Sigma}_{yy}^{\frac{1}{2}} \bar{\mathbf{v}}_t \end{bmatrix},$$

$$\tilde{\mathbf{r}}_t^* = \frac{1}{\sqrt{2}} \begin{bmatrix} \mathbf{\Sigma}_{xx}^{\frac{1}{2}} \tilde{\mathbf{u}}_t^* \\ \mathbf{\Sigma}_{yy}^{\frac{1}{2}} \tilde{\mathbf{v}}_t^* \end{bmatrix}, \qquad \mathbf{r}_t^* = \frac{1}{\sqrt{2}} \begin{bmatrix} \mathbf{\Sigma}_{xx}^{\frac{1}{2}} \mathbf{u}_t^* \\ \mathbf{\Sigma}_{yy}^{\frac{1}{2}} \mathbf{v}_t^* \end{bmatrix}, \qquad \mathbf{r}^* = \frac{1}{\sqrt{2}} \begin{bmatrix} \mathbf{\Sigma}_{xx}^{\frac{1}{2}} \mathbf{u}^* \\ \mathbf{\Sigma}_{yy}^{\frac{1}{2}} \mathbf{v}^* \end{bmatrix}.$$

All these vectors are in $\mathbb{R}^d$ and have length 1.

Observe that the matrix-vector multiplication (8) is equivalent to

$$\begin{bmatrix} \mathbf{\Sigma}_{xx}^{\frac{1}{2}} \tilde{\mathbf{u}}_t \\ \mathbf{\Sigma}_{yy}^{\frac{1}{2}} \tilde{\mathbf{v}}_t \end{bmatrix} \leftarrow \begin{bmatrix} \mathbf{\Sigma}_{xx}^{\frac{1}{2}} & \\ & \mathbf{\Sigma}_{yy}^{\frac{1}{2}} \end{bmatrix} \begin{bmatrix} \lambda\mathbf{\Sigma}_{xx} & -\mathbf{\Sigma}_{xy} \\ -\mathbf{\Sigma}_{xy}^\top & \lambda\mathbf{\Sigma}_{yy} \end{bmatrix}^{-1} \begin{bmatrix} \mathbf{\Sigma}_{xx}^{\frac{1}{2}} & \\ & \mathbf{\Sigma}_{yy}^{\frac{1}{2}} \end{bmatrix} \begin{bmatrix} \mathbf{\Sigma}_{xx}^{\frac{1}{2}} \mathbf{u}_{t-1} \\ \mathbf{\Sigma}_{yy}^{\frac{1}{2}} \mathbf{v}_{t-1} \end{bmatrix},$$

and

$$\begin{bmatrix} \mathbf{\Sigma}_{xx}^{\frac{1}{2}} & \\ & \mathbf{\Sigma}_{yy}^{\frac{1}{2}} \end{bmatrix} \begin{bmatrix} \lambda\mathbf{\Sigma}_{xx} & -\mathbf{\Sigma}_{xy} \\ -\mathbf{\Sigma}_{xy}^\top & \lambda\mathbf{\Sigma}_{yy} \end{bmatrix}^{-1} \begin{bmatrix} \mathbf{\Sigma}_{xx}^{\frac{1}{2}} & \\ & \mathbf{\Sigma}_{yy}^{\frac{1}{2}} \end{bmatrix}$$

$$= \begin{bmatrix} \mathbf{\Sigma}_{xx}^{-\frac{1}{2}} & \\ & \mathbf{\Sigma}_{yy}^{-\frac{1}{2}} \end{bmatrix}^{-1} \begin{bmatrix} \lambda\mathbf{\Sigma}_{xx} & -\mathbf{\Sigma}_{xy} \\ -\mathbf{\Sigma}_{xy}^\top & \lambda\mathbf{\Sigma}_{yy} \end{bmatrix}^{-1} \begin{bmatrix} \mathbf{\Sigma}_{xx}^{-\frac{1}{2}} & \\ & \mathbf{\Sigma}_{yy}^{-\frac{1}{2}} \end{bmatrix}^{-1}$$

$$= \left( \begin{bmatrix} \mathbf{\Sigma}_{xx}^{-\frac{1}{2}} & \\ & \mathbf{\Sigma}_{yy}^{-\frac{1}{2}} \end{bmatrix} \begin{bmatrix} \lambda\mathbf{\Sigma}_{xx} & -\mathbf{\Sigma}_{xy} \\ -\mathbf{\Sigma}_{xy}^\top & \lambda\mathbf{\Sigma}_{yy} \end{bmatrix} \begin{bmatrix} \mathbf{\Sigma}_{xx}^{-\frac{1}{2}} & \\ & \mathbf{\Sigma}_{yy}^{-\frac{1}{2}} \end{bmatrix} \right)^{-1}$$

$$= \begin{bmatrix} \lambda\mathbf{I} & -\mathbf{\Sigma}_{xx}^{-\frac{1}{2}}\mathbf{\Sigma}_{xy}\mathbf{\Sigma}_{yy}^{-\frac{1}{2}} \\ -\mathbf{\Sigma}_{yy}^{-\frac{1}{2}}\mathbf{\Sigma}_{xy}^\top\mathbf{\Sigma}_{xx}^{-\frac{1}{2}} & \lambda\mathbf{I} \end{bmatrix}^{-1}$$

$$= \mathbf{M}_\lambda.$$

Then the updates for exact power iterations can be written as

$$\tilde{\mathbf{r}}_t^* \leftarrow \mathbf{M}_\lambda \mathbf{r}_{t-1}^*, \qquad \mathbf{r}_t^* \leftarrow \tilde{\mathbf{r}}_t^* / \|\tilde{\mathbf{r}}_t^*\|, \qquad t = 1, \ldots,$$

and the updates for inexact power iterations can be written as

$$\tilde{\mathbf{r}}_t \approx \mathbf{M}_\lambda \mathbf{r}_{t-1}, \qquad \mathbf{r}_t \leftarrow \tilde{\mathbf{r}}_t / \|\tilde{\mathbf{r}}_t\|, \qquad t = 1, \ldots.$$

Note we have according to (18) that

$$\tilde{\epsilon} \geq (\tilde{\mathbf{r}}_t - \bar{\mathbf{r}}_t)^\top \mathbf{M}_\lambda^{-1} (\tilde{\mathbf{r}}_t - \bar{\mathbf{r}}_t) \geq \sigma_d(\mathbf{M}_\lambda^{-1}) \cdot \|\tilde{\mathbf{r}}_t - \bar{\mathbf{r}}_t\|^2 = \frac{1}{\sigma_1(\mathbf{M}_\lambda)} \cdot \|\tilde{\mathbf{r}}_t - \bar{\mathbf{r}}_t\|^2$$

or equivalently

$$\|\tilde{\mathbf{r}}_t - \bar{\mathbf{r}}_t\| \leq \sqrt{\sigma_1(\mathbf{M}_\lambda) \cdot \epsilon}. \tag{19}$$

***Proof of Lemma 7.*** Recall that the eigenvectors of $\mathbf{M}_\lambda$ are:

$$\lambda_1 := \frac{1}{\lambda - \rho_1} > \lambda_2 := \frac{1}{\lambda - \rho_2} \geq \cdots \geq \lambda_{d-1} := \frac{1}{\lambda + \rho_2} \geq \lambda_d := \frac{1}{\lambda + \rho_1},$$

with corresponding eigenvectors

$$\mathbf{e}_1 = \mathbf{r}^* = \frac{1}{\sqrt{2}} \begin{bmatrix} \mathbf{a}_1 \\ \mathbf{b}_1 \end{bmatrix}, \ \mathbf{e}_2 = \frac{1}{\sqrt{2}} \begin{bmatrix} \mathbf{a}_2 \\ \mathbf{b}_2 \end{bmatrix}, \ \ldots, \ \mathbf{e}_{d-1} = \frac{1}{\sqrt{2}} \begin{bmatrix} \mathbf{a}_2 \\ -\mathbf{b}_2 \end{bmatrix}, \ \mathbf{e}_d = \frac{1}{\sqrt{2}} \begin{bmatrix} \mathbf{a}_1 \\ -\mathbf{b}_1 \end{bmatrix}.$$

By the update rule of exact power iterations, it holds that for $i = 1, \ldots, d$ that

$$
\begin{aligned}
(\mathbf{e}_i^\top \mathbf{r}_t^*)^2 &= \frac{\left(\mathbf{e}_i^\top \mathbf{M}_\lambda^t \mathbf{r}_0^*\right)^2}{\|\mathbf{M}_\lambda^t \mathbf{r}_0^*\|^2} = \frac{\left(\mathbf{e}_i^\top \mathbf{M}_\lambda^t \mathbf{r}_0\right)^2}{(\mathbf{r}_0^*)^\top \mathbf{M}_\lambda^{2t} \mathbf{r}_0^*} = \frac{\left(\lambda_i^t \mathbf{e}_i^\top \mathbf{r}_0^*\right)^2}{\sum_{j=1}^d \lambda_j^{2t} \left(\mathbf{e}_j^\top \mathbf{r}_0^*\right)^2} = \frac{\left(\mathbf{e}_i^\top \mathbf{r}_0^*\right)^2}{\sum_{j=1}^d \left(\frac{\lambda_j}{\lambda_i}\right)^{2t} \left(\mathbf{e}_j^\top \mathbf{r}_0^*\right)^2} \\
&\leq \frac{\left(\mathbf{e}_i^\top \mathbf{r}_0^*\right)^2}{\left(\frac{\lambda_1}{\lambda_i}\right)^{2t} \left(\mathbf{e}_1^\top \mathbf{r}_0^*\right)^2} = \frac{\left(\mathbf{e}_i^\top \mathbf{r}_0^*\right)^2}{\left(\mathbf{e}_1^\top \mathbf{r}_0^*\right)^2} \left(\frac{\lambda_i}{\lambda_1}\right)^{2t} = \frac{\left(\mathbf{e}_i^\top \mathbf{r}_0^*\right)^2}{\tilde{\mu}} \left(1 - \frac{\lambda_1 - \lambda_i}{\lambda_1}\right)^{2t} \\
&\leq \frac{\left(\mathbf{e}_i^\top \mathbf{r}_0^*\right)^2}{\tilde{\mu}} \cdot \exp\left(-2\frac{\lambda_1 - \lambda_i}{\lambda_1} t\right).
\end{aligned}
$$

Given $\delta \in (0, 1)$, define $S(\delta) = \{i : \lambda_i > (1 - \delta)\lambda_1\}$. For $\delta_1, \delta_2 \in (0, 1)$, define

$$T(\delta_1, \delta_2) := \lceil \frac{1}{2\delta_1} \log\left(\frac{1}{\tilde{\mu}\delta_2}\right) \rceil.$$

For all $i \notin S(\delta_1)$, when $t > T(\delta_1, \delta_2)$, it holds that $(\mathbf{e}_i^\top \mathbf{r}_t^*)^2 \leq \delta_2 (\mathbf{e}_i^\top \mathbf{r}_0^*)^2$, and thus in particular $\sum_{i \in S(\alpha/2)} \left(\mathbf{e}_i^\top \mathbf{r}_t^*\right)^2 \geq 1 - \delta_2$.

Part one (crude regime) of the lemma now follows by noticing that, by setting $\delta_1 = \delta_2 = \frac{\alpha}{2}$ we have that for $t \geq T\left(\frac{\alpha}{2}, \frac{\alpha}{2}\right)$, it holds that

$$(\mathbf{r}_t^*)^\top \mathbf{M}_\lambda \mathbf{r}_t^* = \sum_{i=1}^d \lambda_i \left(\mathbf{e}_i^\top \mathbf{r}_t^*\right)^2 \geq \sum_{i \in S(\alpha/2)} \left(1 - \frac{\alpha}{2}\right) \lambda_1 \left(\mathbf{e}_i^\top \mathbf{r}_t^*\right)^2 \geq \left(1 - \frac{\alpha}{2}\right)^2 \lambda_1 \geq (1 - \alpha)\lambda_1.$$

For the second part (accurate regime) of the lemma, note that $S\left(\frac{\lambda_1 - \lambda_2}{\lambda_1}\right) = \{1\}$. Thus for all $t \geq T\left(\frac{\lambda_1 - \lambda_2}{\lambda_1}, \alpha\right)$, it holds that $(\mathbf{e}_1^\top \mathbf{r}_t^*)^2 \geq 1 - \alpha$.

$\square$

***Proof of Lemma 8.*** We prove the bound for unnormalized iterates by induction. The case for $t = 1$ holds trivially. For $t \geq 2$, we can bound the error of the unnormalized iterates using the exact solution to $\tilde{h}_t$:

$$\|\tilde{\mathbf{r}}_t - \tilde{\mathbf{r}}_t^*\| \leq \|\tilde{\mathbf{r}}_t - \bar{\mathbf{r}}_t\| + \|\bar{\mathbf{r}}_t - \tilde{\mathbf{r}}_t^*\|. \tag{20}$$

The second term of (20) is concerned with the error due to inexact target in the least squares problem $h_t(\mathbf{u}, \mathbf{v})$ as $\begin{bmatrix} \mathbf{u}_{t-1} \\ \mathbf{v}_{t-1} \end{bmatrix}$ is different from $\begin{bmatrix} \mathbf{u}_{t-1}^* \\ \mathbf{v}_{t-1}^* \end{bmatrix}$. We can bound this term as

$$\|\bar{\mathbf{r}}_t - \tilde{\mathbf{r}}_t^*\| = \|\mathbf{M}_\lambda \mathbf{r}_{t-1} - \mathbf{M}_\lambda \mathbf{r}_{t-1}^*\| \leq \|\mathbf{M}_\lambda\| \cdot \|\mathbf{r}_{t-1} - \mathbf{r}_{t-1}^*\|$$
$$= \sigma_1(\mathbf{M}_\lambda) \cdot \|\mathbf{r}_{t-1} - \mathbf{r}_{t-1}^*\|. \tag{21}$$

In view of the update rule of our algorithm and the triangle inequality, we have

$$\|\mathbf{r}_{t-1} - \mathbf{r}_{t-1}^*\|$$
$$\leq \left\| \frac{\tilde{\mathbf{r}}_{t-1}}{\|\tilde{\mathbf{r}}_{t-1}\|} - \frac{\tilde{\mathbf{r}}_{t-1}}{\|\tilde{\mathbf{r}}_{t-1}^*\|} \right\| + \left\| \frac{\tilde{\mathbf{r}}_{t-1}}{\|\tilde{\mathbf{r}}_{t-1}^*\|} - \frac{\tilde{\mathbf{r}}_{t-1}^*}{\|\tilde{\mathbf{r}}_{t-1}^*\|} \right\|$$
$$= \|\tilde{\mathbf{r}}_{t-1}\| \left| \frac{1}{\|\tilde{\mathbf{r}}_{t-1}\|} - \frac{1}{\|\tilde{\mathbf{r}}_{t-1}^*\|} \right| + \frac{1}{\|\tilde{\mathbf{r}}_{t-1}^*\|} \|\tilde{\mathbf{r}}_{t-1} - \tilde{\mathbf{r}}_{t-1}^*\|$$
$$= \frac{1}{\|\tilde{\mathbf{r}}_{t-1}^*\|} \left| \|\tilde{\mathbf{r}}_{t-1}^*\| - \|\tilde{\mathbf{r}}_{t-1}\| \right| + \frac{1}{\|\tilde{\mathbf{r}}_{t-1}^*\|} \|\tilde{\mathbf{r}}_{t-1} - \tilde{\mathbf{r}}_{t-1}^*\|$$
$$\leq \frac{2}{\|\tilde{\mathbf{r}}_{t-1}^*\|} \|\tilde{\mathbf{r}}_{t-1} - \tilde{\mathbf{r}}_{t-1}^*\| \leq \frac{2\tilde{R}_{t-1}}{\|\tilde{\mathbf{r}}_{t-1}^*\|}. \tag{22}$$

For $t \geq 2$, we have $\tilde{\mathbf{r}}_{t-1}^* = \mathbf{M}_\lambda \mathbf{r}_{t-2}^*$ and $\|\mathbf{r}_{t-2}^*\| = 1$, and thus

$$\|\tilde{\mathbf{r}}_{t-1}^*\| \geq \sigma_d(\mathbf{M}_\lambda).$$

Combining (20), (21) and (22) gives

$$\|\tilde{\mathbf{r}}_t - \tilde{\mathbf{r}}_t^*\| \leq \sqrt{\sigma_1(\mathbf{M}_\lambda) \cdot \epsilon} + 2\kappa_\lambda \tilde{R}_{t-1} = \tilde{R}_t.$$

The bound for normalized iterates follows from (22). $\qquad\square$

***Proof of Lemma 9.*** For the first item (crude regime), observe that

$$\mathbf{r}_t^\top \mathbf{M}_\lambda \mathbf{r}_t = (\mathbf{r}_t^*)^\top \mathbf{M}_\lambda \mathbf{r}_t^* + \left( (\mathbf{r}_t^*)^\top \mathbf{M}_\lambda \mathbf{r}_t^* - \mathbf{r}_t^\top \mathbf{M}_\lambda \mathbf{r}_t \right), \tag{23}$$

and that

$$\left| (\mathbf{r}_t^*)^\top \mathbf{M}_\lambda (\mathbf{r}_t^*) - \mathbf{r}_t^\top \mathbf{M}_\lambda \mathbf{r}_t \right| = \left| \left( \mathbf{M}_\lambda^{\frac{1}{2}} \mathbf{r}_t^* + \mathbf{M}_\lambda^{\frac{1}{2}} \mathbf{r}_t \right)^\top \left( \mathbf{M}_\lambda^{\frac{1}{2}} \mathbf{r}_t^* - \mathbf{M}_\lambda^{\frac{1}{2}} \mathbf{r}_t \right) \right|$$
$$\leq \left\| \mathbf{M}_\lambda^{\frac{1}{2}} \mathbf{r}_t^* + \mathbf{M}_\lambda^{\frac{1}{2}} \mathbf{r}_t \right\| \left\| \mathbf{M}_\lambda^{\frac{1}{2}} \mathbf{r}_t^* - \mathbf{M}_\lambda^{\frac{1}{2}} \mathbf{r}_t \right\|$$
$$\leq \left\| \mathbf{M}_\lambda^{\frac{1}{2}} \right\| \|\mathbf{r}_t^* + \mathbf{r}_t\| \left\| \mathbf{M}_\lambda^{\frac{1}{2}} \right\| \|\mathbf{r}_t^* - \mathbf{r}_t\|$$
$$\leq \|\mathbf{M}_\lambda\| \left( \|\mathbf{r}_t^*\| + \|\mathbf{r}_t\| \right) \|\mathbf{r}_t^* - \mathbf{r}_t\|$$
$$= 2\sigma_1(\mathbf{M}_\lambda) \cdot \|\mathbf{r}_t^* - \mathbf{r}_t\|.$$

Our choices of $T$ and $\tilde{\epsilon}$ make sure that $(\mathbf{r}_T^*)^\top \mathbf{M}_\lambda \mathbf{r}_T^* \geq \left(1 - \frac{\alpha}{2}\right) \cdot \sigma_1(\mathbf{M}_\lambda)$ by Lemma 7 and that $\|\mathbf{r}_T^* - \mathbf{r}_T\| \leq R_T = \alpha/4$ by Lemma 8. Continuing from (23), we have

$$\mathbf{r}_T^\top \mathbf{M}_\lambda \mathbf{r}_T \geq \left(1 - \frac{\alpha}{2}\right) \cdot \sigma_1(\mathbf{M}_\lambda) - \frac{\alpha}{2} \cdot \sigma_1(\mathbf{M}_\lambda) = (1 - \alpha) \cdot \sigma_1(\mathbf{M}_\lambda).$$

For the second item (accurate regime), observe that

$$(\mathbf{r}_t^\top \mathbf{r}^*)^2 = \left( (\mathbf{r}_t^*)^\top \mathbf{r}^* + (\mathbf{r}_t - \mathbf{r}_t^*)^\top \mathbf{r}^* \right)^2 \geq \left( (\mathbf{r}_t^*)^\top \mathbf{r}^* \right)^2 - 2 \|\mathbf{r}_t - \mathbf{r}_t^*\|. \tag{24}$$

Our choices of $T$ and $\tilde{\epsilon}$ make sure that $\left( (\mathbf{r}_T^*)^\top \mathbf{r}^* \right)^2 \geq 1 - \frac{\alpha}{2}$ by Lemma 7 and that $\|\mathbf{r}_T^* - \mathbf{r}_T\| \leq R_T = \alpha/4$ by Lemma 8. Continuing from (24), we have

$$(\mathbf{r}_T^\top \mathbf{r}^*)^2 \geq 1 - \frac{\alpha}{2} - \frac{\alpha}{2} = 1 - \alpha.$$

$\qquad\square$

## F.2 Iteration complexity of Algorithm 3

Observe that, the **for** loops within the **repeat-until** loop, as well as the final **for** loop in Algorithm 3 are running inexact power iterations on $\mathbf{M}_{\lambda_{(s)}}$ and $\mathbf{M}_{\lambda_{(f)}}$ for $m_1$ and $m_2$ inexact matrix-vector multiplication respectively. And the convergence of inexact power iterations is provided by Lemma 8.

For each iteration of the **repeat-until** loop, we work in the crude regime and only require $\mathbf{r}_{sm_1}$ to give a constant multiple estimate of $\mathbf{M}_{\lambda_{(s)}}$. The lemma below shows an important property of $\Delta_s$ which is used to locate $\lambda_{(f)}$, and the number of iterations needed to reach $\lambda_{(f)}$.

***Lemma*** 10 (Iteration complexity of the **repeat-until** loop in Algorithm 3). Suppose that $\tilde{\Delta} \in [c_1\Delta,\ c_2\Delta]$ where $c_2 \leq 1$. Set $m_1 = \lceil 8\log\left(\frac{16}{\mu'}\right) \rceil$ and $\tilde{\epsilon} \leq \frac{1}{3084}\left(\frac{\tilde{\Delta}}{18}\right)^{m_1-1}$ in Algorithm 3. Then for all $s \geq 1$ it holds that

$$\frac{1}{2}(\lambda_{(s-1)} - \rho_1) \leq \Delta_s \leq \lambda_{(s-1)} - \rho_1,$$

upon exiting this loop, the $\lambda_{(f)}$ satisfies

$$\rho_1 + \frac{\tilde{\Delta}}{4} \leq \lambda_{(f)} \leq \rho_1 + \frac{3\tilde{\Delta}}{2}, \tag{25}$$

and the number of iterations run by the **repeat-until** loop is $\log\left(\frac{1}{\tilde{\Delta}}\right)$.

*Proof.* Let $\bar{\sigma}$ be an upper bound of all $\sigma_1(\mathbf{M}_{\lambda_{(s)}})$ used in the **repeat-until** loop, i.e.,

$$\bar{\sigma} \geq \sigma_1(\mathbf{M}_{\lambda_{(s)}}), \qquad s = 1, 2, \dots.$$

And suppose for now that throughout the loop, $\tilde{\epsilon}$ satisfies

$$\sqrt{\bar{\sigma}\tilde{\epsilon}} \leq \frac{\sigma_1\left(\mathbf{M}_{\lambda_{(s-1)}}\right)}{8}. \tag{26}$$

Set $\alpha = \frac{1}{4}$ in Lemma 8 (crude regime), and with our choice of $m_1$ and

$$\tilde{\epsilon} \leq \frac{\sigma_d(\mathbf{M}_{\lambda_{(s)}})}{1024\kappa_{\lambda_{(s)}}}\left(\frac{2\kappa_{\lambda_{(s)}} - 1}{\left(2\kappa_{\lambda_{(s)}}\right)^{m_1} - 1}\right)^2, \tag{27}$$

we have

$$\mathbf{r}_{sm_1}^\top \mathbf{M}_{\lambda_{(s-1)}} \mathbf{r}_{sm_1} \geq \frac{3}{4}\sigma_1(\mathbf{M}_{\lambda_{(s-1)}}). \tag{28}$$

In view of the definition of the vector $\mathbf{w}_s$ in Algorithm 3, and following the same argument in (18), we have

$$\left\|\frac{\mathbf{z}_s}{\sqrt{2}} - \mathbf{M}_{\lambda_{(s-1)}}\mathbf{r}_{sm_1}\right\| \leq \sqrt{\sigma_1(\mathbf{M}_{\lambda_{(s-1)}}) \cdot \tilde{\epsilon}}$$

where $\mathbf{z}_s = \begin{bmatrix} \mathbf{\Sigma}_{xx}^{\frac{1}{2}} & \\ & \mathbf{\Sigma}_{yy}^{\frac{1}{2}} \end{bmatrix} \mathbf{w}_s$.

Then for every iteration of the **repeat-until** loop, it holds that

$$\frac{1}{2}\begin{bmatrix} \mathbf{u}_{sm_1}^\top & \mathbf{v}_{sm_1}^\top \end{bmatrix}\begin{bmatrix} \mathbf{\Sigma}_{xx} & \\ & \mathbf{\Sigma}_{yy} \end{bmatrix}\mathbf{w}_s$$

$$= \mathbf{r}_{sm_1}^\top\left(\frac{\mathbf{z}_s}{\sqrt{2}}\right) = \mathbf{r}_{sm_1}^\top \mathbf{M}_{\lambda_{(s-1)}}\mathbf{r}_{sm_1} + \mathbf{r}_{sm_1}^\top\left(\frac{\mathbf{z}_s}{\sqrt{2}} - \mathbf{M}_{\lambda_{(s-1)}}\mathbf{r}_{sm_1}\right)$$

$$\in \left[\mathbf{r}_{sm_1}^\top \mathbf{M}_{\lambda_{(s-1)}}\mathbf{r}_{sm_1} - \sqrt{\sigma_1(\mathbf{M}_{\lambda_{(s-1)}}) \cdot \tilde{\epsilon}},\ \mathbf{r}_{sm_1}^\top \mathbf{M}_{\lambda_{(s-1)}}\mathbf{r}_{sm_1} + \sqrt{\sigma_1(\mathbf{M}_{\lambda_{(s-1)}}) \cdot \tilde{\epsilon}}\right]$$

$$\in \left[\mathbf{r}_{sm_1}^\top \mathbf{M}_{\lambda_{(s-1)}}\mathbf{r}_{sm_1} - \sqrt{\bar{\sigma}\tilde{\epsilon}},\ \mathbf{r}_{sm_1}^\top \mathbf{M}_{\lambda_{(s-1)}}\mathbf{r}_{sm_1} + \sqrt{\bar{\sigma}\tilde{\epsilon}}\right],$$

where we have used the Cauchy-Schwarz inequality in the second step.

In view of (26) and (28), it follows that

$$
\frac{1}{2} \begin{bmatrix} \mathbf{u}_{sm_1}^\top & \mathbf{v}_{sm_1}^\top \end{bmatrix} \begin{bmatrix} \boldsymbol{\Sigma}_{xx} & \\ & \boldsymbol{\Sigma}_{yy} \end{bmatrix} \mathbf{w}_s - \sqrt{\sigma}\tilde{\epsilon}
$$
$$
\in \left[ \mathbf{r}_{sm_1}^\top \mathbf{M}_{\lambda_{(s-1)}} \mathbf{r}_{sm_1} - 2\sqrt{\sigma}\tilde{\epsilon}, \ \mathbf{r}_{sm_1}^\top \mathbf{M}_{\lambda_{(s-1)}} \mathbf{r}_{sm_1} \right]
$$
$$
\in \left[ \frac{1}{2}\sigma_1(\mathbf{M}_{\lambda_{(s-1)}}), \ \sigma_1(\mathbf{M}_{\lambda_{(s-1)}}) \right].
$$

By the definition of $\Delta_s$ in Algorithm 3 and the fact that $\sigma_1(\mathbf{M}_{\lambda_{(s-1)}}) = \frac{1}{\lambda_{(s-1)} - \rho_1}$, we have

$$
\Delta_s = \frac{1}{2} \cdot \frac{1}{\frac{1}{2} \begin{bmatrix} \mathbf{u}_{sm_1}^\top & \mathbf{v}_{sm_1}^\top \end{bmatrix} \begin{bmatrix} \boldsymbol{\Sigma}_{xx} & \\ & \boldsymbol{\Sigma}_{yy} \end{bmatrix} \mathbf{w}_s - \sqrt{\sigma}\tilde{\epsilon}} \in \left[ \frac{1}{2}\left( \lambda_{(s-1)} - \rho_1 \right), \ \lambda_{(s-1)} - \rho_1 \right]. \quad (29)
$$

And as a result,

$$
\lambda_{(s)} = \lambda_{(s-1)} - \frac{\Delta_s}{2} \geq \lambda_{(s-1)} - \frac{1}{2}\left( \lambda_{(s-1)} - \rho_1 \right) = \frac{\lambda_{(s-1)} + \rho_1}{2},
$$

and thus by induction (note $\lambda_{(0)} \geq \rho_1$) we have $\lambda_{(s)} \geq \rho_1$ throughout the **repeat-until** loop.

From (29) we also obtain

$$
\lambda_{(s)} - \rho_1 = \lambda_{(s-1)} - \rho_1 - \frac{\Delta_s}{2} \leq \lambda_{(s-1)} - \rho_1 - \frac{1}{4}\left( \lambda_{(s-1)} - \rho_1 \right) = \frac{3}{4}\left( \lambda_{(s-1)} - \rho_1 \right).
$$

To sum up, $\lambda_{(s)}$ approaches $\rho_1$ from above and the gap between $\lambda_{(s)}$ and $\rho_1$ reduces at the geometric rate of $\frac{3}{4}$. Thus after at most $t_3 = \lceil \log_{3/4}\left( \frac{\tilde{\Delta}}{\lambda_{(0)} - \rho_1} \right) \rceil \sim \mathcal{O}\left( \log\left( \frac{1}{\tilde{\Delta}} \right) \right)$ iterations, we reach a $\lambda_{(t_3)}$ such that $\lambda_{(t_3)} - \rho_1 \leq \tilde{\delta}$. And in view of (29), the **repeat-until** loop exits in the next iteration. Hence, the overall number of iterations is at most $t_3 + 1 = \mathcal{O}\left( \frac{1}{\tilde{\Delta}} \right)$.

We now analyze $\lambda_{(f)}$ and derive the interval it lies in. Note that $\Delta_f \leq \tilde{\Delta}$ and $\Delta_{f-1} > \tilde{\Delta}$ by the exiting condition. In view of (29), we have

$$
\lambda_{(f)} - \rho_1 = \lambda_{(f-1)} - \rho_1 - \frac{\Delta_f}{2} \leq 2\Delta_f - \frac{\Delta_f}{2} = \frac{3\Delta_f}{2} \leq \frac{3\tilde{\Delta}}{2}.
$$

On the other hand,

$$
\lambda_{(f)} - \rho_1 = \lambda_{(f-1)} - \rho_1 - \frac{\Delta_f}{2} \geq \lambda_{(f-1)} - \rho_1 - \frac{1}{2}\left( \lambda_{(f-1)} - \rho_1 \right) = \frac{1}{2}\left( \lambda_{(f-1)} - \rho_1 \right). \quad (30)
$$

If $f = 1$, then by our choice of $\lambda_{(0)}$ we have that $\lambda_{(f)} - \rho_1 \geq \tilde{\Delta}$. Otherwise, by unfolding (30) one more time, we have that

$$
\lambda_{(f)} - \rho_1 \geq \frac{1}{4}\left( \lambda_{(f-2)} - \rho_1 \right) \geq \frac{\Delta_{f-1}}{4} \geq \frac{\tilde{\Delta}}{4}.
$$

Thus in both case, we have that $\lambda_{(f)} - \rho_1 \geq \frac{\tilde{\Delta}}{4}$ holds.

It remains to give an explicit bound on $\tilde{\epsilon}$ based on the two requirements (26) and (27). Since the $\lambda_{(s)}$ values are monotonically non-increasing and lower-bounded by $\rho_1 + \frac{\tilde{\Delta}}{4}$, we have

$$
\max_s \ \sigma_1(\mathbf{M}_{\lambda_{(s)}}) = \sigma_1(\mathbf{M}_{\lambda_{(f)}}) = \frac{1}{\lambda_{(f)} - \rho_1} \leq \frac{4}{\tilde{\Delta}} =: \overline{\sigma},
$$

and

$$
\min_s \ \sigma_1(\mathbf{M}_{\lambda_{(s)}}) = \sigma_1(\mathbf{M}_{\lambda_{(0)}}) = \frac{1}{\lambda_{(0)} - \rho_1} = \frac{1}{1 + \tilde{\Delta} - \rho_1}
$$
$$
\geq \frac{1}{1 + c_2\Delta - \Delta} \geq 1 + (1 - c_2)\Delta \geq 1 + \frac{1 - c_2}{c_2}\tilde{\Delta} := \underline{\sigma},
$$

where the first inequality holds since by definition of $\Delta$ it follows that $\rho_1 = \rho_2 + \Delta \geq \Delta$.

Therefore, for the assumption (26) to hold, we just need

$$\left(\frac{\sigma}{8\sqrt{\bar{\sigma}}}\right)^2 = \frac{\left(1 + \frac{1-c_2}{c_2}\tilde{\Delta}\right)^2}{64 \cdot \frac{4}{\tilde{\Delta}}} \geq \frac{1}{64 \cdot \frac{4}{\tilde{\Delta}}} = \frac{\tilde{\Delta}}{256} \geq \tilde{\epsilon}. \tag{31}$$

We now derive a lower bound of the right hand side of (27). Notice

$$\kappa_{\lambda_{(s)}} = \frac{\lambda_{(s)} + \rho_1}{\lambda_{(s)} - \rho_1} = 1 + \frac{2\rho_1}{\lambda_{(s)} - \rho_1} \leq 1 + 2\rho_1\bar{\sigma} \leq 1 + 2\bar{\sigma} \leq \frac{9}{\tilde{\Delta}}. \tag{32}$$

On the other hand,

$$\sigma_d(\mathbf{M}_{\lambda_{(s)}}) \geq \sigma_d(\mathbf{M}_{\lambda_{(0)}}) = \frac{1}{\lambda_{(0)} + \rho_1} = \frac{1}{1 + \tilde{\Delta} + \rho_1} \geq \frac{1}{3}.$$

As a result, we have

$$\frac{\sigma_d(\mathbf{M}_{\lambda_{(s)}})}{1024\kappa_{\lambda_{(s)}}}\left(\frac{2\kappa_{\lambda_{(s)}} - 1}{\left(2\kappa_{\lambda_{(s)}}\right)^{m_1} - 1}\right)^2 \geq \frac{1}{3084 \cdot \frac{9}{\tilde{\Delta}}}\left(\frac{2\frac{9}{\tilde{\Delta}} - 1}{\left(2\frac{9}{\tilde{\Delta}}\right)^{m_1} - 1}\right)^2 \geq \frac{\left(\frac{17}{\tilde{\Delta}}\right)^2}{3084 \cdot \frac{9}{\tilde{\Delta}} \cdot \left(\frac{18}{\tilde{\Delta}}\right)^{m_1}}$$

$$\geq \frac{1}{3084}\left(\frac{\tilde{\Delta}}{18}\right)^{m_1 - 1}. \tag{33}$$

Our final bound on $\tilde{\epsilon}$ chooses the smaller of (31) and (33). □

For the final **for** loop of Algorithm 3, we work in the accurate regime of power iterations.

***Lemma*** 11 (Iteration complexity of the final **for** loop in Algorithm 3). Suppose that $\tilde{\Delta} \in [c_1\Delta, c_2\Delta]$ where $c_2 \leq 1$. Set $m_2 = \lceil\frac{5}{4}\log\left(\frac{128}{\tilde{\mu}\eta^2}\right)\rceil$ and $\tilde{\epsilon} \leq \frac{\eta^4}{4^{10}}\left(\frac{\tilde{\Delta}}{18}\right)^{m_2-1}$ in Algorithm 3. Then the $(\mathbf{u}_T, \mathbf{v}_T)$ output by Phase I satisfies

$$\frac{1}{4}(\mathbf{u}_T^\top\mathbf{\Sigma}_{xx}\mathbf{u}^* + \mathbf{v}_T^\top\mathbf{\Sigma}_{yy}\mathbf{v}^*)^2 \geq 1 - \frac{\eta^2}{64}. \tag{34}$$

*Proof.* Notice when $\lambda = \rho_1 + c(\rho_1 - \rho_2)$, we have

$$\delta(\mathbf{M}_\lambda) = \frac{\sigma_1(\mathbf{M}_\lambda)}{\sigma_1(\mathbf{M}_\lambda) - \sigma_2(\mathbf{M}_\lambda)} = \frac{\frac{1}{\lambda-\rho_1}}{\frac{1}{\lambda-\rho_1} - \frac{1}{\lambda-\rho_2}} = \frac{\lambda - \rho_2}{\rho_1 - \rho_2} = \frac{\rho_1 + c(\rho_1 - \rho_2) - \rho_2}{\rho_1 - \rho_2} = c + 1.$$

In view of (25), $\lambda_{(f)} - \rho_1 \leq \frac{3}{2}\tilde{\Delta} \leq \frac{3c_2}{2}\Delta \leq \frac{3}{2}\Delta$, and thus $\delta(\mathbf{M}_{\lambda_{(f)}}) \leq \frac{5}{2}$.

Set $\alpha = \frac{\eta^2}{64}$ in Lemma 8 (accurate regime), and with our choice of $m_2$ and

$$\tilde{\epsilon} \leq \frac{\eta^4 \cdot \sigma_d(\mathbf{M}_{\lambda_{(f)}})}{64^3 \cdot \kappa_{\lambda_{(f)}}}\left(\frac{2\kappa_{\lambda_{(f)}} - 1}{\left(2\kappa_{\lambda_{(f)}}\right)^{m_2} - 1}\right)^2, \tag{35}$$

we are guaranteed to obtained the desired alignment.

We now give a lower bound of the right hand side of (35). First,

$$\sigma_d(\mathbf{M}_{\lambda_{(f)}}) = \frac{1}{\lambda_{(f)} + \rho_1} \geq \frac{1}{\rho_1 + \frac{3}{2}\Delta + \rho_1} \geq \frac{1}{4}.$$

Recall that we have proved in (32) that $\kappa_{\lambda_{(f)}} \leq \frac{9}{\tilde{\Delta}}$. Following a derivation similar to that of (33), we have

$$\frac{\eta^4 \cdot \sigma_d(\mathbf{M}_{\lambda_{(f)}})}{64^3 \cdot \kappa_{\lambda_{(f)}}}\left(\frac{2\kappa_{\lambda_{(f)}} - 1}{\left(2\kappa_{\lambda_{(f)}}\right)^{m_2} - 1}\right)^2 \geq \frac{\eta^4}{4^{10}}\left(\frac{\tilde{\Delta}}{18}\right)^{m_2-1}, \tag{36}$$

and this explains the $\epsilon$ we set in the lemma. □

**Proof of Theorem 4.** As shown in Lemma 11, the **repeat-until** loop runs $\mathcal{O}\left(\log\left(\frac{1}{\tilde{\Delta}}\right)\right) \sim$ $\mathcal{O}\left(\log\left(\frac{1}{\Delta}\right)\right)$ iterations, and inside each iteration, we run $m_1$ approximate matrix-vector multiplications. On the other hand, the final **for** loop runs $m_2$ approximate matrix-vector multiplications. By the definitions of $m_1$ and $m_2$, the total number of invocations of approximate matrix-vector multiplications/least squares problems is

$$m_1 \cdot \log\left(\frac{1}{\Delta}\right) + m_2 \sim \mathcal{O}\left(\log\left(\frac{1}{\tilde{\mu}}\right)\log\left(\frac{1}{\Delta}\right) + \log\left(\frac{1}{\tilde{\mu}\eta^2}\right)\right) \sim \tilde{\mathcal{O}}(1).$$

$\square$

## G   Proof of Theorem 5

*Proof.* Notice that the eigenvectors of $\mathbf{M}_\lambda$ form an orthonormal bases of $\mathbb{R}^{d_x+d_y}$. Thus when (34) holds, i.e., the alignment between $\begin{bmatrix} \boldsymbol{\Sigma}_{xx}^{\frac{1}{2}}\tilde{\mathbf{u}}_T \\ \boldsymbol{\Sigma}_{yy}^{\frac{1}{2}}\tilde{\mathbf{v}}_T \end{bmatrix}$ and $\begin{bmatrix} \boldsymbol{\Sigma}_{xx}^{\frac{1}{2}}\mathbf{u}^* \\ \boldsymbol{\Sigma}_{yy}^{\frac{1}{2}}\mathbf{v}^* \end{bmatrix}$ is large, the alignments between $\begin{bmatrix} \boldsymbol{\Sigma}_{xx}^{\frac{1}{2}}\tilde{\mathbf{u}}_T \\ \boldsymbol{\Sigma}_{yy}^{\frac{1}{2}}\tilde{\mathbf{v}}_T \end{bmatrix}$ and other eigenvectors have to be small. In particular, the alignment between $\begin{bmatrix} \boldsymbol{\Sigma}_{xx}^{\frac{1}{2}}\tilde{\mathbf{u}}_T \\ \boldsymbol{\Sigma}_{yy}^{\frac{1}{2}}\tilde{\mathbf{v}}_T \end{bmatrix}$ and the tailing eigenvector $\begin{bmatrix} \boldsymbol{\Sigma}_{xx}^{\frac{1}{2}}\mathbf{u}^* \\ -\boldsymbol{\Sigma}_{yy}^{\frac{1}{2}}\mathbf{v}^* \end{bmatrix}$ has to be small:

$$(\mathbf{u}_T^\top\boldsymbol{\Sigma}_{xx}\mathbf{u}^* - \mathbf{v}_T^\top\boldsymbol{\Sigma}_{yy}\mathbf{v}^*)^2 \leq \frac{\eta^2}{16}. \tag{37}$$

From (37) and (34), we have respectively

$$-\frac{\eta}{4} \leq \left|\mathbf{u}_T^\top\boldsymbol{\Sigma}_{xx}\mathbf{u}^*\right| - \left|\mathbf{v}_T^\top\boldsymbol{\Sigma}_{yy}\mathbf{v}^*\right| \leq \frac{\eta}{4},$$

$$\left|\mathbf{u}_T^\top\boldsymbol{\Sigma}_{xx}\mathbf{u}^*\right| + \left|\mathbf{v}_T^\top\boldsymbol{\Sigma}_{yy}\mathbf{v}^*\right| \geq 2\sqrt{1 - \frac{\eta^2}{64}} \geq 2\left(1 - \frac{\eta}{8}\right)$$

where we have used the fact that $\sqrt{1-x} \geq 1 - \sqrt{x}$ for $x \in [0,1]$ in the second inequality. Averaging the above two inequalities gives

$$\left|\mathbf{u}_T^\top\boldsymbol{\Sigma}_{xx}\mathbf{u}^*\right| \geq 1 - \frac{\eta}{4}, \qquad \left|\mathbf{v}_T^\top\boldsymbol{\Sigma}_{yy}\mathbf{v}^*\right| \geq 1 - \frac{\eta}{4}.$$

Finally,

$$\begin{aligned}
(\hat{\mathbf{u}}^\top\boldsymbol{\Sigma}_{xx}\mathbf{u}^*)^2 + (\hat{\mathbf{v}}^\top\boldsymbol{\Sigma}_{yy}\mathbf{v}^*)^2 &= \frac{(\mathbf{u}_T^\top\boldsymbol{\Sigma}_{xx}\mathbf{u}^*)^2}{\mathbf{u}_T^\top\boldsymbol{\Sigma}_{xx}\mathbf{u}_T} + \frac{(\mathbf{v}_T^\top\boldsymbol{\Sigma}_{yy}\mathbf{v}^*)^2}{\mathbf{v}_T^\top\boldsymbol{\Sigma}_{yy}\mathbf{v}_T} \\
&\geq (1 - \frac{\eta}{4})^2\left(\frac{1}{\mathbf{u}_T^\top\boldsymbol{\Sigma}_{xx}\mathbf{u}_T} + \frac{1}{\mathbf{v}_T^\top\boldsymbol{\Sigma}_{yy}\mathbf{v}_T}\right) \\
&\geq \left(1 - \frac{\eta}{4}\right)^2\frac{4}{\mathbf{u}_T^\top\boldsymbol{\Sigma}_{xx}\mathbf{u}_T + \mathbf{v}_T^\top\boldsymbol{\Sigma}_{yy}\mathbf{v}_T} \\
&\geq 2\left(1 - \frac{\eta}{2}\right) = 2 - \eta
\end{aligned}$$

where we have used the fact that $\frac{1}{x} + \frac{1}{y} \geq \frac{4}{x+y}$ in the first inequality, and (10) in the second inequality. Then the theorem follows from the fact that $(\hat{\mathbf{u}}^\top\boldsymbol{\Sigma}_{xx}\mathbf{u}^*)^2$ and $(\hat{\mathbf{v}}^\top\boldsymbol{\Sigma}_{yy}\mathbf{v}^*)^2$ can be at most 1. $\square$

## H   Condition number of $h_t$ for SVRG

**Lemma** 12. Throughout Algorithm 3, the condition number of $h_t$ for SVRG is at most $\frac{9/c}{\Delta}\tilde{\kappa}$, where

$$\tilde{\kappa} := \frac{\max\limits_i \max\left(\|\mathbf{x}_i\|^2, \|\mathbf{y}_i\|^2\right)}{\min\left(\sigma_{\min}(\boldsymbol{\Sigma}_{xx}), \sigma_{\min}(\boldsymbol{\Sigma}_{yy})\right)}.$$

*Proof.* The gradient Lipschitz constant of $h_t^i(\mathbf{u}, \mathbf{v})$ is bounded by the largest eigenvalue (in absolute value) of its Hessian[8]

$$\mathbf{Q}_\lambda^i = \left[ \begin{array}{cc} \lambda \mathbf{x}_i \mathbf{x}_i^\top & -\mathbf{x}_i \mathbf{y}_i^\top \\ -\mathbf{y}_i \mathbf{x}_i^\top & \lambda \mathbf{y}_i \mathbf{y}_i^\top \end{array} \right],$$

and the largest eigenvalue is defined as

$$\max_{\mathbf{g}_x \in \mathbb{R}^{d_x}, \mathbf{g}_y \mathbb{R}^{d_y}} \beta := \left| [\mathbf{g}_x^\top, \mathbf{g}_y^\top] \mathbf{Q}_\lambda^i \left[ \begin{array}{c} \mathbf{g}_x \\ \mathbf{g}_y \end{array} \right] \right| \qquad \text{s.t.} \quad \|\mathbf{g}_x\|^2 + \|\mathbf{g}_y\|^2 = 1.$$

We have

$$\begin{aligned} \beta &= \left| \lambda (\mathbf{g}_x^\top \mathbf{x}_i)^2 + \lambda (\mathbf{g}_y^\top \mathbf{y}_i)^2 - 2(\mathbf{g}_x^\top \mathbf{x}_i)(\mathbf{g}_y^\top \mathbf{y}_i) \right| \\ &\leq \lambda (\mathbf{g}_x^\top \mathbf{x}_i)^2 + \lambda (\mathbf{g}_y^\top \mathbf{y}_i)^2 + 2 \left| \mathbf{g}_x^\top \mathbf{x}_i \right| \left| \mathbf{g}_y^\top \mathbf{y}_i \right| \\ &\leq \lambda (\mathbf{g}_x^\top \mathbf{x}_i)^2 + \lambda (\mathbf{g}_y^\top \mathbf{y}_i)^2 + (\mathbf{g}_x^\top \mathbf{x}_i)^2 + (\mathbf{g}_y^\top \mathbf{y}_i)^2 \\ &= (\lambda + 1) \left( (\mathbf{g}_x^\top \mathbf{x}_i)^2 + (\mathbf{g}_y^\top \mathbf{y}_i) \right) \\ &\leq (\lambda + 1) \left( \|\mathbf{g}_x\|^2 \|\mathbf{x}_i\|^2 + \|\mathbf{g}_y\|^2 \|\mathbf{y}_i\|^2 \right) \\ &\leq (\lambda + 1) \max \left( \|\mathbf{x}_i\|^2, \|\mathbf{y}_i\|^2 \right) \end{aligned}$$

where we have used the Cauchy-Schwarz inequality and the constraint in the third and the last inequality respectively.

It only remains to bound $\frac{\lambda + 1}{\lambda - \rho}$. Note that we have shown in Lemma 10 that $\lambda \geq \rho_1 + \frac{\tilde{\Delta}}{4}$ throughout Algorithm 3, and thus

$$\frac{\lambda + 1}{\lambda - \rho} = 1 + \frac{1 + \rho}{\lambda - \rho} \leq 1 + \frac{2}{\lambda - \rho} \leq 1 + 2\frac{4}{\tilde{\Delta}} \leq \frac{9}{\tilde{\Delta}} \leq \frac{9/c_1}{\Delta}.$$

$\square$

# I   More details of the experiments

The statistics of these datasets are summaized in Table 2. These datasets have also been used by [3, 4] for demonstrating their stochastic CCA algorithms.

Table 2: Brief summary of datasets.

| Datasets | Description | $d_x$ | $d_y$ | $N$ |
|---|---|---|---|---|
| Mediamill | Image and its labels | 100 | 120 | 30,000 |
| JW11 | Acoustic and articulation measurements | 273 | 112 | 30,000 |
| MNIST | Left and right halves of images | 392 | 392 | 60,000 |

We now provide additional details for the experiments. For s-AppGrad, both gradient and normalization steps are estimated with mini-batches of 100 samples (the authors of [3] suggest that the mini-batch size shall be at least the same magnitude as the dimensionality of the CCA projection). For SI-VR and SI-AVR, within the **repeat-until** loop, we apply SVRG with $M = 2$ epochs to approximately find the top eigenvector $\mathbf{w}_s$, and SVRG with $M = 2$ epochs to approximately calculate its top eigenvalue of $\mathbf{M}_{\lambda_{(s)}}$ as $\mathbf{w}_s^T \mathbf{M}_{\lambda_{(s)}} \mathbf{w}_s$. We exit the **repeat-until** loop when $\Delta_s \leq 0.06$. Afterwards, for the fixed $\lambda_{(f)}$, we apply SVRG to solve every least squares problems with $M = 4$ epochs. Each epoch of SVRG includes a batch gradient evaluation and $m = N$ stochastic gradient steps. We set the step size according to the smoothness for each least squares solver, i.e., $\frac{1}{\sigma_{\max}(\mathbf{\Sigma}_{xx})}$ for GD/AGD in AppGrad/s-AppGrad/CCALin, and $\frac{1}{\max_i \|\mathbf{x}_i\|^2}$ for SVRG/ASVRG in our algorithms.

## J   Other related work

Recent years have witnessed continuous efforts to scale up fundamental methods such as principal component analysis (PCA) and partial least squares with stochastic/online updates [22, 23, 24, 25, 5, 16, 17]. But as pointed out by [23], the CCA objective is more challenging due to the constraints.

[26] proposed an adaptive CCA algorithm with efficient online updates based on matrix manifolds defined by the constraints. However, the goal of their algorithm is anomaly detection for streaming data with a varying distribution, rather than to optimize the CCA objective on a given dataset. Similar to our algorithms, the stochastic CCA algorithms of [3, 4] are motivated by the ALS formulation. [5] proposed a stochastic algorithm based on the Lagrangian formulation of the objective (1). None of these online/stochastic algorithms have rigorous global convergence guarantee.

## Footnotes

[8]We omit the regularization terms, which are typically very small, to have concise expressions.