[Reviews · NeurIPS 2016]

Reviewer 1

Summary

The paper considers a stochastic version in CCA. CCA is a useful tool that, given two snapshots of different set of variables (that come, however, from the same set of subjects), we are interested in revealing the most important correlations between the variables. State of the art algorithms work mostly in an offline fashion: given all datapoints from each set of variables, the task is to solve the maximization of a quadratic bilinear form, over quadratic constraints. While the problem has a "closed-form" solution (after matrix transformations such as matrix inversions), only until recently some online/stochastic methods have been proposed. This paper considers such scenario for CCA and proposes 2 algorithms that solve the online CCA problem with global approximation guarantees. The authors claim that this is the first time such guarantees are provided for such setting. Experimental results show the superiority of the algorithms compared to state of the art.

Qualitative Assessment

Novelty: The novelty of the algorithm lies in providing global convergence guarantees for the online version of CCA. The algorithm has potential - however, this would have been better highlighted if real-applications were used (e.g. neuroscience experiments) Impact/Significance: CCA is an important algorithm that finds applications in Cognitive Neuroscience, NLP, speech recognition, molecular biology, etc. To this end, having an efficient online algorithm for such settings is valuable. Clarity: The authors do a great job presenting exactly the necessary information needed: from introducing CCA to the power method version of CCA, and to its ALS and stochastic formulation. At each step of the description, the authors provide remarks about what can go wrong: what happens when there is a gap, what about the normalization step (is it expensive?), what about input sparsity, etc. The flow of the paper is divided into small paragraphs/subsections, each of which provide an additional piece of information about the proposed algorithms. Really helpful for easy reading. Other comments: 1. The authors should consider comparing also with the recent paper: Raman Arora, Poorya Mianjy and Teodor Marinov. Stochastic optimization for multiview learning using partial least squares. In Proceedings of the 32nd International Conference on Machine Learning (ICML), 2016. In their approach, the authors also consider a (partial) least-squares procedure for solution. 2. It would be helpful if the authors could describe in 1-2 sentences the findings, presented in Table 1. I.e., how much does log^2(1/eta) affect the performance over log(1/eta)? Is ASVRG always better than SVRG? If such overall comments are present, these will be really appreciated by readers. (Some of these comments are scattered over the text - maybe you could consider gathering all the comments together). 3. What is not clear from the text is whether the algorithm 1, which is taken from [7], wasn't proved to converge until now. If this is the first time this algorithm is proved, this should be stressed out in the text. If not, then the proof in the appendix is redundant and Theorem 1 should have a reference. This point is to discussed also with the reviewers, after the rebuttal. 4. How much significant is to set \epsilon(T) as proposed in Theorem 2? It seems that we need to know some information about the problem at hand (singular values, etc)? 5. It is not clear where the upper bound on the sub-optimality of initialization for SVRG is crucial for the proof of Lemma 3, or the overall analytical complexity of Algorithm 2. Can you please elaborate? 6. Maybe a small description of the technique shift-and-invert-preconditioning would be helpful to understand the idea behind it. Currently, the authors directly apply the steps, following similar notions from analogous PCA approaches. Overall, this is a good paper and for the moment I have no reason to push for rejection. However, some re-writing of parts of the paper are needed for publication.

Confidence in this Review

3-Expert (read the paper in detail, know the area, quite certain of my opinion)


Reviewer 2

Summary

This is a well written paper on the numerical resolution of canonical correlation analysis. The author combines advanced linear algebra and stochastic gradient techniques with a strong focus on finite-time complexity estimates.

Qualitative Assessment

I liked the sense of detail of the author but I think that the paper deserves a little more detail (I'd say a full paragraph) on the relation between canonical correlation analysis and machine learning applications. There are a few typos Line 15: trainng -> training Footnote 3: It can show -> One can show Line 13: the the Line 129: ||x_i||^2-smooth -> (\gamma_x + ||x_i||^2)-smooth Line 183: (34) -> (11)

Confidence in this Review

1-Less confident (might not have understood significant parts)


Reviewer 3

Summary

The paper studied stochastic optimization algorithm for canonical correlation analysis. Two algorithms are proposed inspired from previous work. The key observation is to run the intermediate least-squares problem inexactly by stochastic variance reduced gradient algorithms. Another contribution is the global linear convergence and time complexity analysis.

Qualitative Assessment

One of the key contributions of the paper is to establish the global linear convergence of the proposed algorithms. However, the contribution of the theoretical analysis (proof) is not highlighted. On line 113, the authors mentioned that "By the analysis of inexact power iterations ..", it seems to me that the authors are following some existing analysis without giving any citations. Please be specific. A similar ALS algorithms was also proposed in [6]. The authors cited this paper and gave some discussion on the difference of the time complexity. As the reviewer read the referenced paper, the authors in [6] mentioned that only requires black box access to any linear solver, thus SVRG and ASVRG are both applicable. It is recommend to discuss and compare with these results. In particular, is there any difference in terms of algorithm and the theoretical analysis? The second algorithm was built on the shift-and-invert preconditioning method for PCA. It reads to me this part is mostly straightforward extension of their method to the CCA setting. Essentially apply their method to a new matrix to compute the eigenvectors. Is there any significant difference in the analysis? Compared with [6], one shortcoming of this paper is that it only focuses on solving for the leading components, while [6] also can find top-k components. Is it a straightforward extension to finding top-k components? One novelty in this paper seems that the algorithms do not need to solve the least-squares problem exactly.

Confidence in this Review

2-Confident (read it all; understood it all reasonably well)


Reviewer 4

Summary

The authors propose new algorithms (or rather, new meta-algorithms that generates a class of algorithms) for solving the canonical correlation analysis problem. These algorithms solve CCA by transforming it into a sequence of least-squares problems that can be solved approximately. This allows them to leverage work on both power iteration and approximate solutions to least-squares problems to achieve both a rate that is better than their competitors (in terms of the condition number of the problem) and empirical results that seem to outperform their competitors.

Qualitative Assessment

This seems to be an interesting application of this sort of analysis to the particular problem of CCA. The work is explained clearly enough that I both understand what the authors are doing theoretically, and could implement any of the algorithms described in the paper, which is great for an algorithms paper. Additionally, the authors expose a design decision (which internal least-squares algorithm to use) that, as far as I'm aware based on my limited understanding of the area, has not previously been considered; the fact that the choice of approximate least-squares solver has any effect on the overall time complexity of the algorithm is non-obvious. Some questions/comments: 1) What differentiates your algorithm from that of the parallel work of Ge et al? Is your algorithm fundamentally different in some way? Is the difference only in the way in which the algorithm sets its parameters? Is the difference only in the theoretical analysis? If either of the latter two is true, then your rate seems to be off by a bit: can you explain this discrepancy? 2) Can you give some intuition in the experimental section about how fast we can run these algorithms? In order to know which one I should use, I should be able to get a sense of this. Is number of passes a good proxy for total wall clock time? 3) Are there actual problems in which $\tilde \kappa > N$? This seems like a rather unlikely condition, especially for the big problems on which we might be interested in stochastic optimization in the first place.

Confidence in this Review

2-Confident (read it all; understood it all reasonably well)


Reviewer 5

Summary

The paper studies the stochastic optimization of canonical correlation analysis whose objective is nonconvex and does not decouple over training samples. It proposes two globally convergent meta-algorithms for solving alternating least squares and shift-and-invert preconditioning. The theoretical analysis and experimental results demonstrate their superior performance.

Qualitative Assessment

The paper studies CCA problem which is very interesting. The algorithms are explained very clear, and the theorems behind the algorithms are solid and sound. The reviewer believes this is an excellent work and recommends for acceptance.

Confidence in this Review

1-Less confident (might not have understood significant parts)